

# GREB-ISM v0.3: A coupled ice sheet model for the Global Resolved Energy Balance model for global simulations on time-scales of 100 kyr

Zhiang Xie[1, 2], Dietmar Dommenget[1, 2], Felicity S. McCormack[1], Andrew N. Mackintosh[1]

[1] Monash University, School of Earth, Atmosphere and Environment, Clayton, Victoria 3800, Australia
[2] ARC Centre of Excellence for Climate Extremes, Australia

*Correspondence to*: Zhiang Xie (zhiang.xie@monash.edu)

**Abstract.** We introduce a newly developed global ice sheet model coupled to the Globally Resolved Energy Balance (GREB) climate model for the simulation of global ice sheet evolution on time scales of 100 kyr or longer (GREB-ISM v0.3). Ice sheets and ice shelves are simulated on a global grid, fully interacting with the climate simulation of surface temperature,
precipitation, albedo, land-sea mask, topography and sea level. Thus, it is a fully coupled atmosphere, ocean, land and ice sheet model. We test the model in ice sheet stand-alone and fully coupled simulations. The ice sheet model dynamics behave similarly to other hybrid SIA (Shallow Ice Approximation) and SSA (Shallow Shelf Approximation) models, but the West Antarctic Ice Sheet accumulates too much ice using present-day boundary conditions. The coupled model simulations produce global equilibrium ice sheet volumes and calving rates similar to observed for present day boundary conditions. We designed
a series of idealised experiments driven by oscillating solar radiation forcing on periods of 20 kyr, 50 kyr and 100 kyr in the Northern Hemisphere. These simulations show clear interactions between the climate system and ice sheets, resulting in slow build-up and fast decay of ice-covered areas and global ice volume. The results also show that Northern Hemisphere ice sheets respond more strongly to time scales longer than 100 kyr. The coupling to the atmosphere and sea level leads to climate interactions between the Northern and Southern Hemispheres. The model can run global simulations of 100 kyr per day on a
desktop computer, allowing the simulation of the whole Quaternary period (2.6 Myrs) within one month.

## 1 Introduction

Understanding ice-age cycles in the Quaternary period requires an interdisciplinary research approach including the fields of astronomy, geology, physical geography, oceanography and atmospheric science. Geological proxy data show that sea level and surface temperature significantly oscillated with a preferred time scale of about 100 kyr during the last one million years,
indicating that large ice sheets and glaciers formed and retreated many times over this period (Imbrie et al., 1984; Shackelton, 2000; Short et al., 1991). These oscillations in the late Quaternary are known as the ice-age cycles.

By investigating ice-age cycles, researchers have identified many climate processes that generate long-term climate variability. Variations in Earth's orbit and resulting changes in solar forcing have been widely accepted as a major driver of ice age cycles (Imbrie et al., 1984; Milankovitch, 1941; Short et al., 1991; Tabor et al., 2015; Wunsch, 2004). The Earth's axis and variations
in orbital parameters, such as procession, eccentricity and obliquity, can effectively regulate the incoming solar radiation on the Earth surface and season length for both hemispheres, leading to global temperature oscillations on time scales of 20 to 100 kyr (Huybers, 2011; Short et al., 1991). Additionally, greenhouse gases, especially $CO_2$, are considered as a critical forcing





during the late Quaternary (Shackelton, 2000). Before the industrial revolution, atmospheric $CO_2$ varied as an internal climate feedback originating from the ocean, biosphere or lithosphere (Bauska et al., 2018; Hogg, 2008). This carbon cycle of the earth
system significantly changes the surface energy budget and affects climate variability.

The formation of large ice sheets is an important element of climate variability over the last million years (Bintanja and Van De Wal, 2008; Ganopolski and Brovkin, 2017). During ice ages, Northern Hemisphere ice sheets can cover a significant portion of the North American and European continents (Manabe and Broccoli, 1985; Mix et al., 2001), modifying climate through changes in the albedo of snow, low surface temperature, surface elevation and sea level change (Bintanja and Van De
Wal, 2008; Felzer et al., 1996; Hock, 2005; Manabe and Broccoli, 1985; Mix et al., 2001; Overpeck et al., 2006). In addition, there are many other factors that potentially affected the ice age climate system such as deep ocean temperatures, ocean and atmospheric circulation changes, vegetation cover and atmospheric dust content. The interactions between these climate elements led to complex picture during the Quaternary, and the details of these interactions still remain unclear.

Numerical modelling of the ice-climate coupled system is an important way to investigate the effect of ice sheets on the
Quaternary climate system. In the early stage, climate models only simulated the atmosphere and ocean, and ice sheet variations were included as external forcing (Bush, 2004; Gates, 1976; Manabe and Broccoli, 1985; Webb et al., 1998). Most studies with numerical simulations focused on a specific period, like the last glacial maximum, and specific regions, like the Northern Hemisphere (Bush, 2004; Webb et al., 1998) due to limitations in computational resources. Ice sheet modelling often requires the simulation of long (>10 kyr) periods, due to the relatively slow ice sheet adjustment time to climate forcing. Numerical
studies of ice-climate interactions therefore often use decoupled simulations with surface temperature and precipitation taken as boundary conditions for ice sheet models (Greve, 1997; Huybrechts, 2002; Payne et al., 2000). Fortunately, thanks to computer and model developments, progressively more studies apply coupled ice-climate simulations on time-scale of 100 kyr to 1 Myrs (Abe-Ouchi et al., 2013; Ganopolski et al., 2010; Tigchelaar et al., 2019). However, as far as we know, there are currently no global, million year, coupled ice sheet-climate simulations available.

In this study, we introduce a fully coupled ice sheet-climate model as a tool for paleo-climate research. The model is capable of simulating global, coupled ice-climate simulations of 100 kyr within 24 hrs on a desktop computer. It is designed for studies of global interactions between ice sheets and climate on time scales of 100 kyr to 1 Myrs. The starting point for this development is the Globally Resolved Energy Balance (GREB v1.0) climate model, which simulates the fast climate feedbacks relevant for the climate response to external forcing, such as $CO^2$ concentration or variations in solar radiation, on time scales
of up to 500 yrs (Dommenget and Flöter, 2011; Stassen et al., 2019).  We introduce a new ice sheet model into the GREB model, defining the new GREB-ISM model.

The study is organised as follows. First, we introduce the data sets used, followed by the core of the paper which describes the GREB-ISM. This section is organised in three parts: a short introduction of the original GREB model, followed by descriptions of the new ice sheet model and the changes made to the climate simulations in the GREB model to couple the climate system
to the ice sheet model. In section 4 and 5 we present a series of stand-alone ice sheet and fully coupled ice-climate simulations to evaluate the performance of the new model. The final section provides a short summary and discussion.



## 2 Data

Input values for most climatology for the GREB model, such as surface temperature, atmospheric humidity, horizontal winds and vertical air motion, are taken from the ERA-Interim dataset (Dee et al., 2011). Soil moisture is from NCEP reanalysis data from 1950-2008 (Kalnay et al., 1996), cloud cover climatology from the ISCCP project (Rossow and Schiffer, 1991) and ocean mixed layer depth climatology from Lorbacher et al. (2006). Precipitation data is from Global Precipitation Climatology Project (GPCP, Adler et al. 2003) and for Antarctica we use the dataset from NCEP-DOE (Behrangi et al., 2020; Kanamitsu et al., 2002).

The modern observed bed rock and ice thickness data for Greenland and Antarctica are derived from Bedmachine (Morlighem et al., 2017, 2020). In this study, the bed rock refers to all different types of ice basis. Fig. 1 shows the global map in the GREB model resolution of the bed rock and observed ice thickness. Ice sheet calving rates are taken from Bigg (1999) for Greenland and Liu et al. (2015) for Antarctica. For paleoclimate proxies, the Greenland Ice Core Project (GRIP) (Greve, 1997) and Vostok dataset are used to impose surface air temperature anomalies for the last 250 kyr. d[18]O proxy from sea sediment (Imbrie, 1982; Lisiecki and Raymo, 2005) is used as a proxy for global sea level change for the last 250 kyr.

## 3 Model description

We describe the GREB model before we introduce the ice sheet model developed in this study. We discuss changes made to the GREB model to couple the climate variables of the GREB model to the ice sheet variables, introducing the new model: GREB-ISM. All variables of the GREB-ISM model, as discussed in this study, are listed in Table 1. A model schematic of the coupling between the ice sheet and the climate model is illustrated in Fig. 2.

### 3.1 The Globally Resolved Energy Balance (GREB) model

The GREB model is developed and fully described in Dommenget and Flöter (2011), with the additional introduction of a new hydrological cycle model in Stassen et al. (2019). The model has three layers (atmosphere, surface and sub-surface ocean) with a global, horizontal grid spacing of 3.75º x 3.75º (96 x 48 points). The GREB model simulates four prognostic variables: surface ($T_{surf}$), atmospheric ($T_{atmos}$) and subsurface ocean temperature ($T_{ocean}$), and surface humidity ($q_{air}$):

$$\gamma_{surf} \frac{dT_{surf}}{dt} = F_{solar} + F_{thermal} + F_{latent} + F_{sense} + F_{ocean} + F_{correct} \tag{1}$$

$$\gamma_{atmos} \frac{dT_{atmos}}{dt} = -F_{sense} + Fa_{thermal} + Q_{latent} + \gamma_{atmos}(\kappa_a \cdot \nabla^2 T_{atmos} - \vec{u} \cdot \nabla T_{atmos}) \tag{2}$$

$$\frac{dT_{ocean}}{dt} = \frac{1}{\Delta t} \Delta To_{entrain} - \frac{1}{\gamma_{ocean} - \gamma_{surf}} Fo_{sense} + Fo_{correct} \tag{3}$$

$$\frac{dq_{air}}{dt} = \Delta q_{eva} + \Delta q_{precip} + \kappa_a \cdot \nabla^2 q_{air} - \vec{u} \cdot \nabla q_{air} + \Delta q_{correct} \tag{4}$$



The main physical processes that control the surface temperature tendencies are: solar (short-wave) and thermal (long-wave) radiation, the hydrological cycle (including evaporation, moisture transport and precipitation), horizontal transport of heat, and heat uptake in the subsurface ocean. GREB further simulates a number of diagnostic variables, such as precipitation snow/ice
cover and sea ice, resulting from the simulation of the prognostic variables.

Atmospheric circulation (mean winds) and cloud cover are seasonally prescribed boundary conditions, and state-independent flux corrections are used to keep the GREB model close to the observed mean climate. Thus, the GREB model does not simulate the atmospheric or ocean circulation and is therefore conceptually very different from CGCM simulations. The model does simulate important climate feedbacks such as the water vapour and ice-albedo feedback, but an important limitation of
the GREB model is that the response to external forcing or model parameter perturbations do not involve circulation or cloud feedbacks. GREB does not have any internal (natural) variability since daily weather systems are not simulated. Subsequently, the control climate or response to external forcing can be estimated from one single year, assuming an equilibrium has been reached. The primary advantage of the GREB model in the context of this study is its simplicity, speed, and low computational cost. The simulation of one year of global climate with the GREB model can be done about 1sec (about 100,000 simulated
years per day on a desktop computer).

### 3.2 Ice sheet model

The ice sheet model is a global thermomechanical ice flow model that comprises momentum balance, mass balance, and energy balance modules for prognostic variables: thickness and temperature. This subsection will describe the ice sheet model, including the model grid, dynamical methods used, parameterizations and approximations made. The numerical schemes,
which are not fully described in the text due to limited space, are summarized in Table 2.

### 3.2.1 Model grid

The ice sheet model uses the same horizontal grid as the GREB model. The Arakawa C scheme (Pollard and Deconto, 2012) is adopted for the simulations of velocities, with the ice thickness and temperature specified at the centre of the grid, and zonal and meridional velocities are specified at the grid boundary midpoint. For the vertical coordinates, we apply a terrain-following
coordinate, $\xi$, in the ice sheet model, where

$$\xi = \frac{z - \frac{H}{2}}{\frac{H}{2}} \tag{5}$$

Four vertical layers are chosen: the surface layer ($\xi = 1$), two Gaussian nodes ($\xi = \pm\frac{1}{\sqrt{3}}$, nodes for 2 points Gaussian
quadrature, Hildebrand 1987) and the base layer ($\xi = -1$). The vertical integration in the model is based on Gaussian-Jacobi quadrature (Hildebrand, 1987), where temperature vertical distribution is estimated by a polynomial curve fitting according to the four layers, which is expressed by:





$$T(\xi) = c_0 + c_1\xi + c_2\xi^2 + c_3\xi^3 \tag{6}$$

where $T$ is the temperature, $c_i$ ($i = 0,1,2,3$) are regression coefficients derived from the temperatures at the above four vertical
nodes at each time step. The global, horizontal model grid has cyclic boundary conditions. For the grid points at the poles, we
assume the poleward neighbour is the point at the same latitude, but shifted by 180º, following the approach in Allen et al.
(1991). To avoid numerical instability in the polar regions, a zonal wave filter is applied from 76.875ºS to the South Pole (Lin
and Rood, 1997; Suarez and Takacs, 1995).

**3.2.2 Glacier mask**

The GREB-ISM ice sheet evolution depends on whether the ice is grounded (land), floating (ice shelves) or if we have thin ice
over the ocean (sea ice). In detail:

- ***Grounded ice (land) points***: ice sheet is grounded on bed rock, satisfying the condition (Larour et al. 2012):

  $b + \frac{\rho_i}{\rho_o}H > 0.$

- ***Floating ice (ice shelves) points***: ice thickness $H \geq 10$m and does not reach the bed rock, satisfying the floating

  condition: $b + \frac{\rho_i}{\rho_o}H \leq 0.$

- ***Ocean points***: all other points. The ocean points here include sea ice grid ($H \geq 0$) as well.

**3.2.3 Mass balance**

The ice surface elevation, calculated from the mass balance equation, is the primary input from the ice sheet model to the
GREB model, calculated for all global grid points. The mass balance equation is:

$$\frac{\partial H}{\partial t} = s - a - \nabla \cdot (\vec{V}_m H) \tag{7}$$

where the accumulation of snow ($s$), ablation (melting) of ice ($a$), and ice transport ($\nabla \cdot (\vec{V}_m H)$) control the mass balance. The
methods used to calculate the terms on the right hand side depend on whether ice is grounded (ice sheet), floating (ice shelves),
or sea ice.  The mass balance for sea ice is described in Subsection 3.3.4. For the ice sheet and shelves, the two local surface
forcing terms for the ice mass balance from equation (7) are the source (accumulation) and sink (ablation) terms. The
accumulation is due to snowfall:

$s = \frac{\rho_o}{\rho_i} r \cdot p \tag{8}$





with the snowfall ratio, **r**:

$$r = \begin{cases} 1, & T_{atmos} < T_m \text{ and } T_{surf} < T_m - 2^oC \\ \frac{1}{2}\left(1 - \frac{T_{surf}-T_m}{2^oC}\right), & T_{atmos} < T_m \text{ and } T_m - 2^oC < T_{surf} < T_m + 2^oC \\ 0, & otherwise \end{cases} \tag{9}$$

The ice ablation rate is due to surface melting by positive surface heat flux:

$$a = -\frac{F_{ice}}{\rho_i L_m} \tag{10}$$

With the latent heat flux for melting ice, $F_{ice}$:


$$F_{ice} = \begin{cases} F_{surf} & \text{partial melting:} \quad -F_{surf} < Fmax_{melt} \\ \frac{\rho_i L_m H}{\Delta t} & \text{complete melting:} -F_{surf} > Fmax_{melt} \\ 0 & \text{no melting:} \qquad F_{surf} > 0 \end{cases} \tag{11}$$

Here, the maximum heat flux for complete ice melting is $Fmax_{melt} = \rho_i L_{ice} \frac{H}{\Delta t}$. The surface heat flux, $F_{surf}$, only considers the net surface heat flux beyond the freezing point:


$$F_{surf} = \gamma_{surf} \frac{T_{se}-T_m}{\Delta t} \tag{12}$$

and the estimated surface temperature without ice fusion is:

$$T_{se} = T_0 + \Delta t \frac{F_{net}}{\gamma_{surf}} \tag{13}$$

The snow accumulation and melting, as described above, control all land ice and snow cover, and therefore also simulate the seasonal cycle of snow and ice cover over land. Fig. 3 illustrates the seasonal cycle of ice thickness in both hemispheres as simulated by the GREB-ISM model with present day boundary conditions. The ice thickness change for ocean points comes

from sea ice changes, which is described in Subsection 3.3.4. Overall, both ice extent and thickness describe the seasonal cycle of ice distribution well.





A boundary condition for the mass transport equations is required at the ice front: here, ice from the ice sheet can be freely advected to the attached ocean grid and become sea ice. In this way, calving is diagnosed as transport from ground (land) or floating ice (shelves) onto ocean points.

### 3.2.4 Momentum balance

Ice flow on grounded elements is governed by the shallow ice approximation (SIA; Hutter 1983; Morland 1984) to the full Stokes equations for momentum balance:

$$\vec{V} = \vec{V}_b - 2\rho_i g \nabla z_{topo} \int_{z_b}^{z} A \exp\left(\frac{-Q}{RT}\right) \sigma_e^{n-1} (H - z') dz' \tag{14}$$

$$\vec{V}_m = \frac{1}{z - z_b} \int_{z_b}^{z} \vec{V} \, dz' \tag{15}$$

and the shallow shelf approximation (SSA; Macayeal 1989) on floating elements:

$$\frac{\partial}{r_e \cos\phi \, \partial\lambda}\left(\eta_{SSA} H (4\frac{\partial V_x}{r_e \cos\phi \, \partial\lambda} + 2\frac{\partial V_y}{r_e \, \partial\phi})\right) + \frac{\partial}{r_e \cos\phi \, \partial\phi}\left(\eta_{SSA} H \left(\frac{\partial V_x}{r_e \, \partial\phi} + \frac{\partial V_y}{r_e \cos\phi \, \partial\lambda}\right) \cos\phi\right) = \rho_i g H \frac{\partial z_{topo}}{r_e \cos\phi \, \partial\lambda} \tag{16}$$

$$\frac{\partial}{r_e \cos\phi \, \partial\phi}\left(\eta_{SSA} H (4\frac{\partial V_y}{r_e \, \partial\phi} + 2\frac{\partial V_x}{r_e \cos\phi \, \partial\lambda})\cos\phi\right) + \frac{\partial}{r_e \cos\phi \, \partial\lambda}\left(\eta_{SSA} H \left(\frac{\partial V_x}{r_e \, \partial\phi} + \frac{\partial V_y}{r_e \cos\phi \, \partial\lambda}\right)\right) = \rho_i g H \frac{\partial z_{topo}}{r_e \, \partial\phi} \tag{17}$$

Vertical velocities are recovered through incompressibility:

$$w = - \int_{z_b}^{z} \nabla \cdot \vec{V} \, dz \tag{18}$$

The deformation of ice under stress is described by Glen's flow law (Glen, 1953, 1954, 1955):

$$\eta = \frac{1}{2EA\sigma_e^{n-1}}, \qquad A = A_0 \exp\left(\frac{-Q}{RT'}\right) \tag{19}$$

Where T' is the temperature corrected for the dependence of melting point on pressure:

$$T' = T - \beta (H - z) \tag{20}$$

In our model, the viscosity $\eta_{SSA}$ has been set as a constant value to match with the observed ice surface velocity and calving in the standalone dynamic equilibrium experiment (Subsection 4.3). Each of equations (14)-(18) above are expressed in z-coordinates, but are transformed into $\xi$-coordinates for the model integration. Boundary conditions for the mechanical model are required at the ice sheet surface, base, and at the ice shelf-ocean front. A stress-free ice surface is assumed:





$$\boldsymbol{\sigma} \cdot \boldsymbol{n} = \boldsymbol{0}$$ (21)

where $\boldsymbol{n}$ is the normal unit vector at the ice surface. At the base, the horizontal ice velocities follow the viscous-type sliding law defined in Greve (1997):

$$\vec{V}_b = -C_{sl} H ||\nabla z_{topo}||^2 \nabla z_{topo}, \quad z = z_b$$ (22)

The stress conditions for the horizontal ice shelf velocities at the interface with the open ocean points follow Greve and Blatter (2009), which in our model is expressed as:


$$4 \frac{\partial}{r_e \cos\phi\, \partial\lambda} \left( \eta_{SSA} H \frac{\partial v_x}{r_e \cos\phi\, \partial\lambda} \right) + 2 \frac{\partial}{r_e \cos\phi\, \partial\lambda} \left( \eta_{SSA} H \frac{\partial v_y}{r_e\, \partial\phi} \right) = \rho_i g H \frac{\partial z_{topo}}{r_e \cos\phi\, \partial\lambda}$$ (23)

$$4 \frac{\partial}{r_e \cos\phi\, \partial\phi} \left( \eta_{SSA} H \frac{\partial v_y}{a\, \partial\phi} \cos\phi \right) + 2 \frac{\partial}{r_e \cos\phi\, \partial\phi} \left( \eta_{SSA} H \frac{\partial v_x}{r_e \cos\phi\, \partial\lambda} \cos\phi \right) = \rho_i g H \frac{\partial z_{topo}}{r_e\, \partial\phi}$$ (24)

### 3.2.5 Energy balance

The ice temperature (energy) balance:

$$\frac{\partial T}{\partial t} = -\vec{V} \cdot \nabla T - w \frac{\partial}{\partial z} T + \frac{\partial}{\partial z} \frac{\kappa}{\rho_i C_p} \frac{\partial}{\partial z} T + \frac{1}{\rho_i C_p} \left( \sigma_{xz}, \sigma_{yz} \right) \cdot \frac{\partial \vec{V}}{\partial z}$$ (25)

The ice temperature balance at the surface is constrained by $T_{surf}$ as computed in the GREB-ISM model (see Subsection 3.3.1):

$$T = T_{surf}, \quad z = z_{topo}$$ (26)

The bottom layer geothermal flux is set as a constant value as in Huybrechts et al. (1996) and Payne et al. (2000):

$$\frac{\partial T}{\partial z} = -\frac{\rho_i C_p G}{\kappa}, \quad z = z_b$$

### 235 3.3 Coupling of the GREB model to the ice sheet

The introduction of an ice sheet model requires a number of changes to the original GREB model. In the following, we describe the changes made to the GREB model equations and illustrate how they affect the simulation of the GREB climate.





### 3.3.1 Energy exchange between GREB and ice sheet / sea ice

The introduction of a prognostic ice sheet model introduces the additional heat flux term, $F_{ice}$ for the $T_{surf}$ tendency eq. (1),
resulting in the new equation:

$$\gamma_{surf} \frac{dT_{surf}}{dt} = F_{solar} + F_{thermal} + F_{latent} + F_{sense} + F_{ice} + F_{ocean} + F_{correct} \qquad (28)$$

The calculations of $F_{ice}$ are described in the mass balance Subsection 3.2.3 and the sea ice Subsection 3.3.4. The effect of $F_{ice}$
can best be illustrated by a simple response experiment, in which we add a 10 m ice cover and evaluate how surface temperature
responds to it (Fig. 4). In this response experiment 10 m of ice cover is introduced over a large region of Europe (Fig. 4d, black
box) at the start of the simulation and then the fully-coupled GREB-ISM model is run to respond to this change.
The introduction of the ice cover forces surface temperature below the freezing point at all locations, as long as the ice sheet
is present (Fig. 4a-c). The atmospheric heat fluxes and sea ice dynamics force the sea ice to melt, which it does faster over the
ocean points due to horizontal sea ice transport. Over land the ice cover melts after the first year and allow surface temperature
to go back to the control run values. The atmospheric heat and moisture transport cause cooling in adjacent regions (Fig. 4d).

### 3.3.2 Surface heat capacity

The surface layer effective heat capacity ($\gamma_{surf}$) in the GREB model is equal to the heat capacity of a water column of the
mixed layer depth over ice free ocean points and equivalent to 2 m soil for all other points (e.g. land and ice covered). Thus,
the formation of sea ice changes the heat capacity from that of the mixed layer depth to a 2 m soil column. This is unchanged
from the original GREB model.

### 3.3.3 Precipitation correction

The hydrological cycle model in GREB developed in Stassen et al. (2019) aimed at a realistic simulation of precipitation with
a focus on the regions of greatest precipitation, i.e. the tropical oceans. While the precipitation model is very good in these
regions (Stassen et al., 2019), it only has limited skills over higher latitude land regions, which are most important for the ice
sheet mass balance of the GREB-ISM.
To allow the ice sheet mass balance to receive unbiased mean precipitation forcing under present day conditions, we introduced
a land precipitation correction in the GREB-ISM model. The new precipitation equation with flux correction is expressed as:

$$\Delta q_{precip} = \Delta q_{precip_{S2019}} + q_{zonal} \cdot p_{correct} \qquad (29)$$





where $\Delta q_{precip_{S2019}}$ is precipitation derived from equation (11) in Stassen et al. (2019) and $q_{zonal} \cdot p_{correct}$ is the flux correction of the equation. The flux corrections are only active over land and are a function of calendar month. They are estimated in a way that the simulated $\Delta q_{precip}$ matches the precipitation data in Section 2 for every calendar month of the year.

Here, we note that the $\Delta q_{precip_{S2019}}$ model assumes that precipitation is proportional to the local humidity ($q_{air}$). Stassen et al. (2019) demonstrate that this assumption is less appropriate in higher latitude land regions, as there is no clear relationship between the local $q_{air}$ and $\Delta q_{precip}$. We therefore set the correction term to be proportional to the zonal mean $q_{air}$ defining $q_{zonal}$, assuming that precipitation over higher latitude land is approximately linearly related to the zonal mean $q_{air}$. Within 30° of the poles $q_{zonal}$ is estimated as the mean from the pole to 60°.

With this approach the precipitation over higher latitude land responds to cooling or warming similarly to other regions (e.g. oceans for lower latitudes). We will discuss the precipitation response of the GREB-ISM further below in the context of the response experiments.

### 3.3.4 Sea ice

Sea ice is a diagnostic variable in the original GREB model, but is now changed to be a prognostic variable in GREB-ISM.

Over land and ice shelf points, ice thicknesses (H) follow the dynamics described in the ice sheet model Subsection 3.2. Over ocean points we use the same prognostic variable (H), but the sea ice thickness dynamics follow a different tendency equation, namely:

$$\frac{\partial H}{\partial t} = \Delta H_{seaice} - \kappa_{si} \nabla^2 H \tag{30}$$

with the local sea ice growth:

$$\Delta H_{seaice} = \frac{-F_{ice}}{\rho_i L_m} \tag{31}$$

and where the latent heat of ice fusion $F_{surf}$ is defined by eqs. (11-13):

$$
\begin{array}{llll}
F_{ice} = F_{surf} & \text{ice grows:} & T_{se} < T_{sm}, F_{surf} < 0 \ and \ H < 0.5 \ m & \\
F_{ice} \ from \ equation \ (11) & \text{ice melts:} & T_{se} > T_{sm}, F_{surf} > 0 & (32) \\
F_{ice} = 0 & \text{no change:} & \text{otherwise} &
\end{array}
$$

Sea ice transport is estimated by isotropic diffusion ($\kappa_{si} \nabla^2 H$). This approximates the effect of turbulent winds and ocean currents transporting sea ice, leading to fast decay of sea ice near open ocean. The diffusion coefficient $\kappa_{si}$ was chosen to roughly lead to a sea ice decaying time scale of about one month.





### 3.3.5 Albedo coupled to ice sheet

The surface albedo ($\alpha_{surf}$) in the original GREB model was diagnosed as function of $T_{surf}$, but is now diagnosed as a function

of the ice thickness ($H$):

$$
\begin{aligned}
\alpha_{surf} &= 0.1 & H &= 0.0 \\
\alpha_{surf} &= 0.1 + 17.5 \text{ m}^{-1} \cdot H & H &\in [0.0,\ 0.02\ \text{m}] \\
\alpha_{surf} &= 0.45 & H &> 0.02\ \text{m}
\end{aligned}
\tag{33}
$$


The linear relation between ice thickness and albedo in the GREB-ISM model was estimated from the assumption that for the observed Northern Hemispheric seasonal cycle of snow/ice cover over land the overall albedo matches the mean overall albedo of the original GREB model.

### 3.3.6 Topography coupled to ice sheet

The land topography ($z_{topo}$) in the original GREB model is a fixed boundary condition that influences a number of processes: thermal radiation, hydrological cycle and the transport of heat and moisture by advection and diffusion. For GREB-ISM the land topography is now a function of the bed rock and ice sheet height:

$$
\begin{aligned}
z_{topo} &= b + H, & \text{for grounded ice} \\
z_{topo} &= \left(1 - \frac{\rho_i}{\rho_o}\right)H, & \text{for floating ice}
\end{aligned}
\tag{34}
$$


The GREB-ISM does not simulate any glacial isostatic adjustment.

### 3.3.7 Sensible heat flux between surface and atmosphere

The variable land topography ($z_{topo}$) should affect the sensible heat flux between $T_{surf}$ and $T_{atmos}$, which was not simulated

in the original GREB model. Here it needs to be considered that the GREB model does not resolve the vertical structure of the atmosphere, as it only has one atmospheric layer. However, in the real world $T_{atmos}$ decreases with surface elevation, following a moist adiabatic lapse rate. We therefore change the sensible heat flux between $T_{surf}$ and $T_{atmos}$, which was approximated in the original GREB model by Newtonian coupling between $T_{surf}$ and $T_{atmos}$. In the GREB-ISM model this is now replaced with a Newtonian coupling between $T_{surf}$ and an adjusted $T_{atmos}$:


$$
F_{sense} = ct_{sense}\left(T_{atmos} + \Gamma \cdot z_{topo} - T_{surf}\right)
\tag{35}
$$





Here we choose a globally constant moist adiabatic lapse rate $\Gamma = -6\ \mathrm{K\ km^{-1}}$. The effect of this sensible heat flux is illustrated with a simple response experiment, see Fig. 5. For this experiment we increase $z_{topo}$ by 1000 m over the centre of Asia, and
show the response of the annual mean $T_{surf}$ and precipitation relative to a control simulation with no changes in $z_{topo}$ (Fig. 5). $T_{surf}$ decreases in response to the topographic perturbation, approximately linearly to the moist adiabatic lapse rate. The higher topography also affects the hydrological cycle, reducing the precipitation locally and also remotely through transport of relatively reduced atmospheric humidity.

### 3.3.8 Sea level and land-sea mask

A sea level subroutine is added in GREB-ISM. Only grounded ice thickness change impacts the global sea level. Consequently, the sea level change $slv$ is defined by:

$$slv = \frac{\int_{grounded}(\mathrm{H-H_{ref}})dA}{\mathrm{A_{ocean}}} \tag{36}$$

where $H_{ref}$ is the reference ice thickness, $A_{ocean}$ is total area of ocean grid and $\int_{grounded} dA$ is an integration over all grounded ice points. $slv$ will be added to bed rock elevation $b$, which eventually impacts the land-sea mask. The sea level and land-sea mask are updated every model year.

The soil moisture, which is a boundary condition for estimating surface evaporation is initially set to observed values over land and then changes if land-sea distribution alters. If the sea level lowers and an ocean point turns into a land point (b > 0)
then the land point has a soil moisture value of 0.3. In turn, if the sea level rises and a land point turns into an ocean point (b < 0), then the soil moisture value is set to 1.0.

### 3.3.9 Meridional heat transport

The study by Dommenget et al. (2019) showed that the GREB model, without flux corrections for $T_{surf}$, has a high latitude climate that is too cold and a tropical climate that is too warm, indicating that the meridional heat transport is too weak. The
meridional heat transport in the GREB model results from the atmospheric heat transport by the mean advection due to the mean horizontal wind field and by isotropic diffusion. The latter depends on the diffusion coefficient $\kappa_a = 8 \times 10^5\ m^2\ s^{-1}$ in the GREB model. This value is not strongly constrained by observations and may effectively be different by an order of magnitude. Since the meridional heat transport may play an important role in the global ice age cycle, we enhance this diffusion coefficient by a factor of 5. This reduces the mean $T_{surf}$ bias in higher latitudes and the tropics in the GREB model without
flux corrections, while at the same time does not increase biases in other locations, indicating it is a better approximation of the isotropic diffusion.

## 4 Model benchmark: Ice sheet model stand-alone simulations





We start our evaluation of the new ice sheet model GREB-ISM with standalone ice sheet model simulations forced with idealized or observed boundary conditions. These simulations focus on the ice sheet simulation only. Subsections 4.1 and 4.2

use standard experiments from the European Ice Sheet Modelling Initiative (EISMINT) model intercomparison Phase I (Huybrechts et al., 1996) and II (Payne et al., 2000). These benchmark experiments test the ice sheet model response to idealised mass and temperature forcing within a given horizontal resolution, with the ice mechanics decoupled from the thermodynamics in EISMINT I and coupled in EISMINT II. In subsection 4.3, we discuss a simulation on the global GREB-ISM grid forced with observed boundary conditions to estimate the dynamically-forced equilibrium of the ice sheet model.

Finally, we discuss an idealised time-varying ice sheet response experiment, based on temperature and precipitation from geological proxy data over the past 250 kyr.

### 4.1 EISMINT I

All simulations in EISMINT I (Huybrechts et al. 1996, H96 hereafter) are based on a regional grid in Cartesian coordinates, while the GREB-ISM ice sheet model is based on a global spherical coordinate grid around the south pole. We therefore

changed the GREB-ISM grid for these experiments to a model grid with 96 points in the zonal and 144 points in the meridional direction (3.75˚ x 1.25˚). Only the first 15 points in the meridional direction are used for the ice sheet simulation. The ice sheet divide in these simulations is the south pole and the length of the meridional grid is 50 km. The simulations are integrated for 200 kyr, but near equilibrium is reached after about 50 kyr.

The mass balance $S$ and surface temperature $T_{surf}$ forcings are given as:

Fixed margin experiment:
$$\begin{cases} S = 0.3 \text{ m yr}^{-1} \\ T_{surf} = T_{min} + S_T d^3 \end{cases} \tag{37}$$

Moving margin experiment:
$$\begin{cases} S = \min\{S_{max}, S_b(R_{el} - d)\} \\ T_{surf} = (270 \text{ K} - S_H H) \end{cases} \tag{38}$$

where $d$ is the distance from south pole and the parameters $S_T$, $T_{min}$, $S_{max}$, $S_b$ and $R_{el}$ are set to $8 \times 10^{-8}$ K km$^{-3}$, 239 K, 0.5 m yr$^{-1}$, 0.01 m yr$^{-1}$km$^{-1}$ and 450 km respectively. Table 3 shows the comparison between the new ice sheet model GREB-

ISM and model results from H96. The GREB-ISM simulations of the ice thickness at divide, and mass flux at midpoint are similar to those found in H96 for both the fixed and moving margin experiments.

The transition experiments with oscillating forcing of temperature and mass balance with periods of 20 kyr and 40 kyr are presented in Fig. 6. The GREB-ISM ice thickness simulation is similar to those of H96 from both fixed and moving margin experiments. In both experiments, the basal temperature at the divide is about one to two degrees colder than in the H96

simulations, which may relate to our coarse vertical resolution.

### 4.2 EISMINT II



EISMINT II experiments (Payne et al. 2000, P2000 here after) involve coupling between the mechanical and thermodynamical components of the ice sheet model. These experiments are designed to test how the ice sheet temperature variations interact with the ice sheet transport. The GREB-ISM model grid used is similar as in EISMINT I, but the number of points in the meridional direction is increased from 15 to 31 and the length of the meridional grid is set to 25 km. All experiments are integrated for 200 kyr. The boundary conditions for the first experiment (**A**) are:

$$
\begin{cases}
S = min\{S_{max}, S_b(R_{el} - d)\} \\
\quad T_s = T_{min} + S_T d
\end{cases}
\tag{39}
$$

where $d$ = 25 km. Parameter $S_T$, $T_{min}$, $S_{max}$, $S_b$ and $R_{el}$ are set as $1.67 \times 10^{-2}$ K km$^{-1}$, 238.15 K, 0.5 m yr$^{-1}$, 0.01 m yr$^{-1}$ km$^{-1}$ and 450 km respectively.

The results of experiment A are summarised in Table 4. The final GREB-ISM values for ice volume, area, divide thickness and basal temperature at the ice sheet divide are all within the range of the models in P2000, indicating a fairly good agreement. The basal melt fraction is underestimated by the GREB-ISM by about 30%, which is possibly related to a cold bias at the bed of the ice sheet. Since the focus in our analysis is mainly in ice thickness, the bias in basal temperature and melting will be discussed in the future.

Experiment B and C in EISMINT II are designed for testing the model sensitivity to various boundary conditions. $T_{min}$ in experiment B is set as 5 K lower than in experiment A, to evaluate the sensitivity of the model to the mean ice temperature. Table 4 depicts the difference between experiment B and A. The GREB-ISM shows, in general, similar changes in ice volume, ice divide thickness, and ice divide basal temperature as in P2000. However, the basal melt fraction change shows a significant discrepancy, which may be due to the cold bias of the basal temperature in experiment A.

For experiment C, $S_{max}$ and $R_{el}$ are set as 0.25 m yr$^{-1}$ and 425 km respectively to evaluate the impact of different mass balances. The results of experiment C are shown in Table 4. For the changes in ice volume, area, divide thickness and divide basal temperature, the response difference between Experiment C and A in GREB-ISM is roughly equivalent to results from P2000. The changes in melt fraction in the GREB-ISM deviate from those of P2000, which is again likely to be related to the cold bias in basal temperatures in the GREB-ISM in experiment A.

Overall, the model reproduces the total ice thickness and ice cover well in the idealised experiments of EISMINT I and II. Although there is a bias in the basal temperature estimation in GREB-ISM, this issue does not have a significant impact on the ice thickness and cover area, which suggests the model is appropriate for global climate and ice evolution simulations.

## 4.3 Globally forced dynamical equilibrium

We now focus on simulating the observed global ice sheets forced with present-day boundary conditions. Although we cannot assume that observed Greenland and Antarctic Ice Sheets are in equilibrium with present day forcing, the dynamic equilibrium simulation should produce a global ice sheet distribution similar to the current observations.





Ice surface temperature and precipitation forcings in the experiment are set to the climatologies derived from ERA-interim,
NCEP-DOE and GPCP data. GREB-ISM is run for 200 kyr, initialized with observed ice thickness. Figures 7-9 show results
from this simulation and Table 5 compares the simulation values of total ice volume boundary calving with observed values
from the literature.

The model reaches an equilibrium after about 50 kyr for both the Northern and Southern Hemispheres. Greenland ice
thicknesses and calving rates show only small differences compared with the initial values. They are also within the estimated
calving values from observation (Bigg, 1999). The difference trends in Antarctica are larger, in particular over West Antarctica.
Here we see a significant increase in ice volume and calving (Fig. 7d and 9b). The West Antarctic ice sheet thickness increase
appears to be inconsistent with the observed values, suggesting a model limitation.

We could not find the specific limitation that is causing this bias. The precipitation forcing does play a role in controlling the
West Antarctic Ice Sheet, but we could not find any reasonable precipitation forcing that would result in significantly better
simulations of the West Antarctic Ice Sheet. The parameterization of the floating ice for ice shelves (SSA) also impacts the
simulation of West Antarctic Ice Sheet. The ice shelf can grow and become grounded as an ice sheet with lower viscosity.
However, again we could not find any reasonable value for the ice viscosity ($\eta_{SSA}$) that would significantly reduce this bias.

The simulated ice surface velocity for Antarctica and Greenland shows a reasonable pattern, capturing the main features of the
transport (Fig. 8). For Antarctica the ice flow is slow in the interior and faster near the boundaries. The largest velocities (more
than 1000 m yr$^{-1}$) appear in ice shelf regions (Ross and Filchner-Ronne Ice Shelf), which is due to the parameterization of the
floating ice for ice shelves (SSA). All of these features are similar to Antarctic surface velocity observations (Mouginot et al.
2012). Greenland ice velocities are also in good agreement with observations (Joughin et al., 2010).

### 4.4 Transition experiment

A further test of the GREB-ISM ice sheet model is a transition experiment with time-varying boundary conditions. This will
evaluate the capability of the global ice sheet model to respond to realistic changes in the boundary conditions. We therefore
design an experiment similar to the one discussed in Greve (1997) for the Greenland Ice Sheet, but extend the expermental
design to the whole globe to evaluate the response of the ice sheet on a global scale.

The surface temperature and precipitation forcings for this experiment are:

$$T_{surf}(\lambda, \phi, t) = T^{today}(\lambda, \phi, t_{day}) + \Delta T_{ma}(\phi, t_{yr})$$

$$(40)$$

$$S(\lambda, \phi, t) = S^{today}(\lambda, \phi, t_{day}) \left(1 + \frac{\Delta T_{ma}(\phi, t_{yr})}{20^oC}\right)$$

The surface temperature ($T_{surf}$) and ice mass balance (S) are present-day regional and seasonally varying climatologies
($T^{today}$, $S^{today}$) plus an annual mean forcing term for $T_{surf}$ and S that varies only with latitude according to $\Delta T_{ma}$. Here, $\Delta T_{ma}$
is the annual mean temperature anomalies compared with present-day (i.e. 2000 AD) derived from the Greenland Ice Core



Project (GRIP) dataset for the Northern Hemisphere and Vostok dataset for the Southern Hemisphere (both data came from SICOPOLIS input data, Greve 1997)). From 45° S to 45° N $\Delta T_{ma}$ linearly changes from Southern to Northern Hemispheric values. The simulation is integrated between -250 kyr to present and initialized with present-day observed ice thickness.

The time series in Fig. 10 depicts the sea level change in this simulation from -200 kyr BP compared with a d[18]O proxy timeseries from sea sediments (Imbrie et al., 1984). The two curves show similar time series variations with a correlation of 0.64. This indicates that qualitatively the GREB-ISM ice sheet shows similar overall global ice sheet variations to those observed over the past 200 kyr. However, the GREB-ISM sea level varies by about 50 m, whereas observations suggest sea level changes are in the order of 120 m (Fairbanks, 1989; Lambeck et al., 2014), indicating that the simulated ice sheet volume

variations are smaller than observed. The sea level is also 20 m lower than present day due to the excess West Antarctic Ice Sheet volume that we also observed in the dynamical equilibrium simulation.

There are several significant extremes in the past 200 kyr simulation, which correspond to the Last Interglacial (LIG; -127 kyr), Last Glacial Maximum (LGM; -21 kyr) and present day. The ice sheet thicknesses for these three time periods are shown in Fig. 11. During the EIVM, only the Greenland Ice Sheet thickness exceeded 400 m in the Northern Hemisphere and the

Antarctic Ice Sheet thickness is similar to present day. During the LGM a series of glaciers are thicker than one thousand meters in Asia and North America. However, ice sheet growth at the LGM is concentrated around the Arctic ocean coastal perimeter, and large European (e.g. Fennoscandia) and North American (Laurentide) ice sheets do not grow as they should (Clark et al., 2009; Velichko et al., 1997).

The overall estimate of ice sheet volume in Greenland and Antarctica for the Last Interglacial, Last Glacial Maximum and

Late Holocene from GREB-ISM and from Fyke et al. (2010) are presented in Table 6. Overall, our simulation the Greenland Ice Sheet is similar to Fyke et al. (2010) but with larger time variations. However, the simulation of Antarctica ice thickness shows very little to no variations between these three periods with very different climate states.

Some of the limitations of the GREB-ISM model in reproducing the observed past ice sheet variations maybe due to the highly simplified experimental setup, such as simplified Tsurf forcing. It is beyond this study to fully explore these deviations, but

the results indicate that the GREB-ISM ice sheet model does have realistic responses to time varying boundary conditions.

## 5 Model benchmark: GREB-ISM coupled simulations

We now focus on the fully coupled GREB-ISM model, in which the ice sheet and other climate variables are interacting in both directions. In the following sections, two sets of experiments are presented. First a dynamic equilibrium experiment is conducted, which is similar to the experiment discussed in Subsection 4.3, but now fully coupled with fixed boundary

conditions. Second, a set of experiments with shortwave radiation oscillating on periods of 20 kyr, 50 kyr and 100 kyr for the Northern Hemisphere are conducted. Those two experiments are designed to evaluate how coupling influences the model's behaviour and to what extent the ice sheet responds to periodic solar forcing. The discussion of these experiments will focus on the introduction of the GREB-ISM model. A more detailed analysis of the ice sheet dynamics coupled with climate dynamics is left for future studies.





## 5.1 Dynamic equilibrium for present day conditions

In this experiment, the GREB-ISM model is fully coupled and forced with the fixed boundary conditions of present-day 340 ppm CO2 concentration and solar radiation. $T_{surf}$ and land precipitation are flux corrected to the mean present-day values. However, those flux corrected variables can respond to changes in the climate system, since the flux correction terms are state-independent (see Subsection 3.1). The simulation is 200 kyr long and results are shown in Figs. 12 and 13.

$T_{surf}$ and precipitation show no long term drift and are close to the observation (Fig. 12a, c). Both reach equilibrium after about 50 kyr. The global ice volume difference is mainly contributed by ice thickness difference in Southern Hemisphere (Fig. 12b), which is similar to the one in the forced experiment discussed in subsection 4.3 (Figs. 7 and 9). As the ice volume increases, the sea level shows a clear decrease tendency and reach equilibrium after 50 kyr as well. The ice thickness spatial pattern in coupled experiment is comparable to the standalone experiment (Figs. 13 and 9). Overall, this control run simulation shows that the coupled GREB-ISM system converges towards an equilibrium state close to the observed one. The simulated trends appear to be mostly due to the anomalous growth of the West Antarctic Ice Sheet.

## 5.2 Shortwave radiation oscillation experiment

In the following experiments we use the same set up as in the previous section, but allow the Northern Hemisphere shortwave radiation, *sw*, to oscillate, taking the form:

$$sw(t) = \left(1 + A_{sw} \cdot sin\left(2\pi \frac{t}{pd}\right)\right) \cdot sw_{present} \tag{41}$$

where $A_{sw}$ is the amplitude of the *sw* oscillations, which increases from 0 at 13° N to 0.1 at 35° N and maintains 0.1 northward of 35° N. The oscillation period, *pd*, is set to 20 kyr, 50 kyr and 100 kyr in three individual simulations. The *sw* oscillation is relative to the present-day solar radiation, $sw_{present}$. The shortwave maximum amplitude is about 20 W m⁻² at 65° N in the annual mean (Fig. 14a-c) and varies with latitudes and seasons (not shown). The 20 kyr, 50 kyr and 100 kyr oscillation periods are simulated for 210, 325 and 350 kyr. The time series for selected climate variables are shown in Fig. 14. The results are shown in reference to the final year of the control run, which is the coupled dynamical equilibrium simulation in Subsection 5.1. To illustrate ice form and retreat in one cycle, we show results from the last forcing cycle of each simulation in Figs. 15-17.

Starting with the 20 kyr oscillation run, there are a number of interesting aspects to point out (Figs. 15a, d, 16a, d and 17a-d). First, at the initial half cycle, the ice volume is slightly lower than the reference state, indicating a warming period leads to deglaciation (Figs 14a-c). Then, after second cycle, the ice volume is always larger than in the control simulation and the cycles are very similar to each other. If we focus on the last cycle of the simulations (Figs. 15-17), we note that $T_{surf}$ and precipitation are mostly in phase with each other and with the shortwave radiation forcing. The Northern Hemispheric $T_{surf}$ oscillation amplitude is about +/- 6 °C and the mean value is clearly below zero (the control run value). This is despite the fact that the





mean shortwave radiation is the same as in the control run. This suggests that the oscillating shortwave radiation has a mean cooling effect. This overall cooling is related to the overall increase in the mean ice sheet volume and extent.

The ice sheet response to the 20 kyr shortwave oscillation has a number of interesting aspects. As mentioned above, the mean ice sheet volume is larger than in the control run. Indeed, it is never smaller than in the control run, not even at the minimum (compare Fig. 13c and 16a), with the exception for the first cycle. Ice sheet area and volume are out of phase. The ice sheet area grows first and is nearly 180° out-of-phase with the SW forcing. The ice sheet volume lags behind the ice sheet extent and reaches its maximum nearly 90° (a quarter cycle) after the minimum in shortwave radiation (Fig. 15a). This illustrates that the ice sheets have not had enough time to equilibrate with the *sw* forcing. Further, we can notice that the ice sheet growth and decay is asymmetric, with a slower build up and faster decay in ice volume, with the reverse pattern in ice sheet area. In the build-up phase the ice sheet extends over large regions at lower latitudes, but has relatively thin ice (Fig. 16b). In the decaying phase the ice sheets retreat to higher latitudes and the ice sheet is relatively thick (Fig. 16d).

The Northern Hemispheric *sw* forcing also leads to a response in the Southern Hemisphere climate (Fig. 15d). This is mainly due to the GREB-ISM atmospheric heat and moisture transport. It is also partly due to the change in global sea level induced by the Northern Hemispheric ice sheet changes. The Southern Hemisphere ice sheet changes are in-phase with the Northern Hemisphere climate. It is further noted that the amplitude of the Southern Hemisphere precipitation response relative to $T_{surf}$ is bigger than in the Northern Hemisphere (compare Fig. 15a and d; given the same scaling factors). This suggests that the moisture transport is more affected by the Northern Hemispheric climate change than the heat transport.

The longer 50 kyr and 100 kyr period runs show a number of changes relative to the 20 kyr run. First, the ice sheet volume amplitudes increase relative to the 20 kyr run, illustrating that the ice sheets are more sensitive to longer time period forcings (Fig. 15a-c). Second, we see a shift of the maximum ice volume closer to the phase of the minimum of the *sw* forcing, suggesting that the ice sheets become closer to equilibrium with longer period *sw* forcing. However, even the 100 kyr oscillation run still shows a significant delay in the ice sheet volume extrema relative to the forcing extrema, indicating that the ice sheets are not yet in equilibrium with the forcings. This illustrates that the intrinsic time scales of the Northern Hemispheric ice sheets are longer than 100 kyr. It is further interesting to note that the ice sheets can extend over shallow oceanic regions, like the Hudson Bay, Bering Strait or Artic Sea in the Siberian sector (Fig. 16g, k), but at the same time do not extend into deep ocean regions (compare Fig. 1c with Fig. 16g, k).

The increase in ice thickness response for the longer 50 kyr and 100 kyr period runs has, however, little impact on the amplitudes of the $T_{surf}$, precipitation and ice cover response in the Northern Hemisphere, which also occurs in the Southern Hemisphere (Fig. 15e and f). For ice sheet in the Southern Hemisphere, the ice thickness is almost keeping constant, which indicates the Antarctica Ice Sheet in the GREB-ISM is not very sensitive to the orbital forcing in Northern Hemisphere.

## 6 Summary and discussion

In this study we introduced a newly developed global ice sheet model coupled to the GREB model, defining the new model GREB-ISM. The ice sheet is simulated on the global grid fully interacting with the climate simulation on all grid points. The





ice sheet mass balance is driven by accumulation of snow, melting by surface heat fluxes and changes due to ice transport. The ice transport follows the shallow ice approximation for grounded ice and shallow shelf approximation for ice shelves. Sea ice-climate interactions are also included.

The GREB-ISM climate simulation interacts with ice sheets through surface temperature, precipitation, albedo, land-sea mask, topography and sea level. To allow for these interactions, the original GREB model was changed by: improving the

precipitation simulation of land, including a prognostic sea ice thickness scheme, coupling the surface albedo to the ice thickness, allowing variable land topography as function of ice thickness, introducing global sea level variation and associated changes in land-sea masks and improving the meridional turbulent, atmospheric heat transport. Thus, the new GREB-ISM is a fully coupled atmosphere, ocean, land and ice sheet model.

We evaluated the performance of the stand-alone ice sheet model in a series of idealized and realistic ice sheet model

simulations. We conducted simulations following the EISMINT I and II idealized experiments and found that the GREB-ISM ice sheet model performs similarly to other models with some limitations in the simulation of internal ice temperature.  In simulations with realistic climate forcing close to present-day, we found that the equilibrium Greenland and most of the East Antarctic ice thickness distribution is very similar to observed, but the West Antarctic Ice Sheet gains too much ice. The overall surface ice velocities and associated calving rates of this model are similar to those observed for both Greenland and East

Antarctica.

We investigated the West Antarctic Ice Sheet thickness bias, by evaluating whether uncertainties in precipitation and the parameterisation of the ice shelf dynamics could cause this bias. However, we found that this bias is unlikely to be caused by these limitations alone and it is likely to also result from other, so far unknown, limitations in the GREB-ISM model. The coarse grid resolution of this model, for instance, may play a role in this limitation (Cuzzone et al., 2019).

A time dependent-simulation with simplified surface temperature and precipitation forcing of the past 250 kyr illustrated that the ISM-GREB model can produce a realistic ice sheet response for Greenland. However, while ISM-GREB grows some ice in the Northern Hemisphere, it does not produce realistic North American or Fennoscandian ice sheets, which may relate to the absence of ocean circulation (Risebrobakken et al., 2007). The results for the Antarctic Ice Sheet are less conclusive, but may be due to the simplified setup of the experiment.

We further conducted a series of coupled GREB-ISM simulations to evaluate the full interaction of all climate elements in the model. The coupled model simulations produce global equilibrium ice sheets and calving rates very similar to observed for present-day boundary conditions. Much of this success in creating a realistic global ice sheet is related to the fact that the GREB-ISM model works with flux correction of surface temperature and land precipitation. This leads to realistic mass balance estimates for the ice sheets even in a fully interactive coupled simulation.

When forced with idealized, oscillating solar radiation forcing on the Northern Hemisphere with different oscillation periods (20 kyr, 50 kyr and 100 kyr) the model responds with growth of large continental ice sheets and clear interactions with the climate system in the Northern and Southern Hemispheres. The simulations illustrated asymmetries in the build-up and decay of large ice sheets in response to periodic forcing, showing that the ice sheets are more sensitive to longer time scales forcings. These experiments illustrate the potential of this model for exploring such interactions in future studies.

In summary, we presented a new model that is suited for the simulations of global-scale climate variability on time scales of 100 kyr and longer. The model is computationally efficient, calculating 100,000 model years global simulations per day on a desktop computer, allowing the simulation of the whole Quaternary period (2.6 Myrs) within one month. For simulations of climate and ice sheet variability over the Quaternary period the GREB-ISM model is, as presented here, a good starting point. Further development may include other relevant climate processes, such as the carbon cycle, deep ocean reservoirs or the

ability of the atmosphere and ocean circulation to respond to changes in topography and the climate state, as well as glacial isostatic adjustment. Such further developments are possible within the framework of the GREB-ISM model and will be addressed in future studies.

## 7 Code availability

The GREB-ISM source code, the model input data as well as a simple user manual are available on Zenodo:
https://zenodo.org/badge/latestdoi/372993505. The reader can redo the simulations in the paper by following the instruction from README.md. The model license is Creative Commons Attribution 4.0 International.

## 8 Author contributions

Zhiang Xie developed the new ice sheet model code and together with Dietmar Dommenget design all benchmark experiments. Dietmar Dommenget helped modify the model coupling process. Felicity S. McCormack and Andrew N. Mackintosh provided
ice sheet related data and useful comments on the ice sheet dynamics and simulation.

## 9 Competing interests

The authors declare that they have no conflict of interest.

## 10 Acknowledgements

This study was supported by the Australian Research Council (ARC) Centre of Excellence for Climate Extremes (CLEX). We
gratefully acknowledge Dr. Ralf Greve of Hokkaido University and Dr. Ed Bueler of University of Alaska Fairbanks on their suggestions for ice sheet dynamics and modelling. We also thanks Dr. Chen-shuo Fan of Monash University and Dr. Tao Han of Institute of Earth Environment, CAS for useful discussion.



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



Table 1: Symbol and parameters list for the GREB-ISM model.

| variable name | symbol | dimensions | value/unit |
|---|---|---|---|
| ice sheet softness parameter | $A$ | t | Pa m$^{-3}$ |
| softness parameter in isotherm case | $A_0$ | constant | $1.96 \times 10^3$ Pa m$^{-3}$ $(T' > -10\text{℃})$<br>$3.99 \times 10^{-13}$ Pa m$^{-3}$ $(T' < -10\text{℃})$ |
| ocean area | $A_{ocean}$ | t | $m^2$ |
| ablation rate | $a$ | x, y, t | m s$^{-1}$ |
| bed rock elevation | $b$ | x, y, t | m |
| regression coefficient for ice temperature | $c_i$ for $i$ from 0 to 3 | x, y, t | K |
| sensible heat bulk coefficient | ct$_{sense}$ | constant | $22.5\ W\ m^{-2}\ K^{-1}$ |
| specific heat capacity for ice | $C_p$ | constant | 2009 J kg$^{-1}$ K$^{-1}$ |
| slide law coefficient for basal velocity | $C_{sl}$ | constant | $6 \times 10^4 yr^{-1}$ |
| enhance factor for SIA | $E$ | constant | 3 |
| air thermal heat | $Fa_{thermal}$ | x, y, t | W m$^{-2}$ |
| surface flux correction | $F_{correct}$ | x, y, t | W m$^{-2}$ |
| ice latent heat | $F_{ice}$ | x, y, t | W m$^{-2}$ |
| latent heat | $F_{latent}$ | x, y, t | W m$^{-2}$ |
| total energy for melting all ice | $Fmax_{melt}$ | x, y, t | W m$^{-2}$ |
| net energy without ice latent heat | $F_{net}$ | x, y, t | W m$^{-2}$ |
| land-sea heat difference | $F_{ocean}$ | x, y, t | W m$^{-2}$ |
| ocean heat flux correction | $Fo_{correct}$ | x, y, t | W m$^{-2}$ |
| sensible heat between ocean and surface | $Fo_{sense}$ | x, y, t | W m$^{-2}$ |
| sensible heat between air and surface | $F_{sense}$ | x, y, t | W m$^{-2}$ |
| solar radiation | $F_{solar}$ | x, y, t | W m$^{-2}$ |
| surface net heat flux without ice | $F_{surf}$ | x, y, t | W m$^{-2}$ |
| net longwave radiation on surface | $F_{thermal}$ | x, y, t | W m$^{-2}$ |
| geothermal heat flux | $G$ | constant | $4.2 \times 10^{-2}$ W m$^{-2}$ |
| ice thickness | $H$ | x, y, t | m |
| ice thickness reference for 0 sea level | $H_{ref}$ | x, y, t | m |
| latent heat of fusion | $L_m$ | constant | $3.335 \times 10^5$ J kg$^{-1}$ |
| precipitation | $p$ | x, y, t | $m\ s^{-1}$ |
| precipitation correction | $p_{correct}$ | x, y, t | kg kg$^{-1}$ s$^{-1}$ |
| activate energy | $Q$ | constant | $1.39 \times 10^5 (T' > -10\text{℃})$<br>$6.4 \times 10^4 (T' < -10\text{℃})$ |
| latent heat in air | $Q_{latent}$ | x, y, t | W m$^{-2}$ |
| air specific humidity | $q_{air}$ | x, y, t | kg kg$^{-1}$ |
| zonal specific humidity mean | $q_{zonal}$ | x, y, t | kg kg$^{-1}$ |
| universal gas constant | $R$ | constant | 8.314 J mol$^{-1}$ K$^{-1}$ |
| snowfall rate | $r$ | x, y, t | unitless |
| Earth radius | $r_e$ | constant | $6.37 \times 10^6 m$ |
| ice accumulation rate (snowfall) | $s$ | x, y, t | m s$^{-1}$ |
| sea level | $slv$ | t | m |
| ice strata temperature | $T$ | x, y, z, t | K |
| homologous temperature corrected by pressure melting point | $T'$ | x, y, z, t | K |





| | | | |
|---|---|---|---|
| air temperature | $T_{atmos}$ | x, y, t | K |
| ice melting temperature | $T_m$ | x, y, z, t | K |
| ocean temperature | $T_{ocean}$ | x, y, t | K |
| estimated temperature without ice latent heat | $T_{se}$ | x, y | K |
| sea water frozen temperature | $T_{sm}$ | constant | $271.45K$ |
| surface temperature | $T_{surf}$ | x, y, t | K |
| ice vertical velocity | w | x, y, z, t | m s$^{-1}$ |
| wind velocity at 850hPa | $\vec{u}$ | x, y | m s$^{-1}$ |
| ice flow horizontal velocity (strata) | $\vec{V}$ | x, y, z, t | m s$^{-1}$ |
| ice flow horizontal velocity (base) | $\vec{V}_b$ | x, y, t | m s$^{-1}$ |
| ice flow horizontal velocity (vertical mean) | $\vec{V}_m$ | x, y, t | m s$^{-1}$ |
| surface velocity zonal component for ice shelf | $V_x$ | x, y, t | m s$^{-1}$ |
| surface velocity meridian component for ice shelf | $V_y$ | x, y, t | m s$^{-1}$ |
| ice flow horizontal velocity (vertical mean) | $\vec{V}_m$ | x, y, t | m s$^{-1}$ |
| altitude above sea level | z | z | $m$ |
| ice sheet bottom layer | $z_b$ | x, y, t | $m$ |
| surface topography | $z_{topo}$ | x, y, t | m |
| surface albedo | $\alpha_{surf}$ | x, y, t | unitless |
| Clausius–Clapeyron gradient | $\beta$ | constant | $8.7\times10^{-4}K\ m^{-1}$ |
| lapse rate | $\Gamma$ | constant | $-0.006\ K\ m^{-1}$ |
| heat capacity of atmosphere layer | $\gamma_{atmos}$ | x, y, t | J K$^{-1}$ m$^{-2}$ |
| heat capacity of ocean layer | $\gamma_{ocean}$ | x, y, t | J K$^{-1}$ m$^{-2}$ |
| heat capacity of surface layer | $\gamma_{surf}$ | x, y, t | J K$^{-1}$ m$^{-2}$ |
| humidity tendency due to precipitation | $\Delta q_{precip}$ | x, y, t | kg kg$^{-1}$ s$^{-1}$ |
| humidity tendency due to correction | $\Delta q_{correct}$ | x, y, t | kg kg$^{-1}$ s$^{-1}$ |
| humidity tendency due to evaporation | $\Delta q_{eva}$ | x, y, t | kg kg$^{-1}$ s$^{-1}$ |
| humidity tendency due to precipitation | $\Delta q_{precip}$ | x, y, t | kg kg$^{-1}$ s$^{-1}$ |
| sea ice mass balance | $\Delta H_{seaice}$ | x, y, t | $m\ s^{-1}$ |
| ocean temperature tendency due to entertainment | $\Delta To_{entrain}$ | x, y, t | K |
| model time step (GREB) | $\Delta t$ | constant | 12 hrs |
| ice viscosity | $\eta$ | t | Pa s |
| ice viscosity for ice shelf | $\eta_{SSA}$ | constant | $2\times10^{14}$ Pa s |
| ice sheet diffusion coefficient | $\kappa$ | constant | 2.1 W (K m)$^{-1}$ |
| air diffusion rate | $\kappa_a$ | constant | $4\times10^6$ m$^2$ s$^{-1}$ |
| sea ice diffusion rate | $\kappa_{si}$ | constant | 0.25 m$^2$ month$^{-1}$ |
| longitude | $\lambda$ | x | degree |
| ice sheet model vertical coordinate | $\xi$ | z | 1 |
| ice density | $\rho_i$ | constant | 910 kg m$^{-3}$ |
| ocean density | $\rho_o$ | constant | 991 kg m$^{-3}$ |
| stress tensor | $\sigma$ | x, y, t | N m$^{-2}$ |
| stress tensor component at a-b direction | $\sigma_{ab}$ | x, y, t | N m$^{-2}$ |



| | | | |
|---|---|---|---|
| effective stress | $\sigma_e$ | t | N m$^{-2}$ |
| latitude | $\phi$ | y | degree |





**Table 2.** Process and its relevant numeric scheme for the ice sheet model.

| Process | Time step | Contribute to | Scheme |
|---|---|---|---|
| Mass balance | half day (GREB) | ice thickness | energy balance |
| Advection | one year | ice thickness | finite volume (FFSL, Lin and Rood 1996) |
| Vertical diffusion | one year | ice temperature | finite difference |
| Vertical advection | one year | ice temperature | finite difference |
| Deformation heat | one year | ice temperature | on vertical sheer of horizontal velocity |





**Table 3.** EISMINT I steady state experiment result comparison between GREB-ISM and the model ensemble from H96. F and M in the experiment description represent fixed-margin and moving-margin experiments, respectively.

| Experiment | ice thickness at divide m | Mass flux at midpoint $10^2$ m$^2$a$^{-1}$ | Basal temperature at divide °C |
|---|---|---|---|
| EISMINT I (F) | $3384.4 \pm 39.4$ | $794.99 \pm 5.67$ | $-8.97 \pm 0.71$ |
| GREB-ISM (F) | 3399.06 | 750.14 | -11.74 |
| EISMINT I (M) | $2978.0 \pm 19.3$ | $999.38 \pm 23.55$ | $-13.34 \pm 0.56$ |
| GREB-ISM (M) | 2916.025 | 1234.40 | -14.93 |





**Table 4.** Results for basic glaciological quantities in EISMINT II experiments after 200 kyr. The percentage in volume, melt fraction and divide thickness column are defined as the difference between two experiments divided by its counterpart in experiment A. The results of P2000 are shown in the form of "mean ± range". See text for details.

| Exp. A | volume $10^6$ km$^3$ | area $10^6$ km$^3$ | Melt fraction | Divide thickness m | Divide basal temperature K |
|---|---|---|---|---|---|
| GREB-ISM | 2.065 | 0.932 | 0.466 | 3829.77 | 254.038 |
| P2000 | 2.128 ±0.145 | 1.034 ±0.086 | 0.719 ±0.290 | 3688.342 ±96.740 | 255.605 ±2.929 |
| Model (Exp. label) | volume % | area % | Melt fraction % | Divide thickness % | Divide basal temperature K |
| GREB-ISM (B) | -4.066 | / | 38.642 | -5.821 | 4.576 |
| P2000 (B) | -2.589 ±1.002 | / | 11.836 ±18.669 | -4.927 ±1.316 | 4.623 ±0.518 |
| GREB-ISM (C) | -25.907 | -17.079 | -100 | -12.137 | 3.856 |
| P2000 (C) | -28.505 ±1.204 | -19.515 ±3.554 | -27.806 ±31.371 | -12.928 ±1.501 | 3.707 ±0.615 |





**Table 5.** Ice volume and boundary calving from the forced dynamic equilibrium experiment and observation.

| Experiment(region) | total ice volume $10^6 km^3$ | boundary calving $10^{12}$ kg |
|---|---|---|
| Observation (Greenland) | 2.83 (Greve, 1997) 3.12 (BedMachine) | 170 - 270 (Bigg et al., 1999) |
| GREB ISM (Greenland) | 3.36 | 211.91 |
| Observation (Antarctica) | 25.6 (Martin et al., 2011) 26.8 (BedMachine) | $1781 \pm 64$ (Liu et al., 2015) |
| GREB ISM (Antarctica) | 32.04 | 2231.83 |





**Table 6.** Annual mean ice volume in the stand-alone transition experiment for different time periods the from GREB-ISM simulation and from Fyke et al. (2010).

| Scenario | GREB-ISM Greenland $10^6$ km$^3$ | Fyke et al. (2010) Greenland $10^6$ km$^3$ | GREB-ISM Antarctica $10^6$ km$^3$ | Fyke et al. (2010) Antarctica $10^6$ km$^3$ |
|---|---|---|---|---|
| LIG | 1.92 | 2.19 | 32.77 | 31.2 |
| LGM | 4.23 | 3.69 | 31.18 | 40.4 |
| Late Holocene | 3.56 | 3.47 | 32.49 | 30.9 |

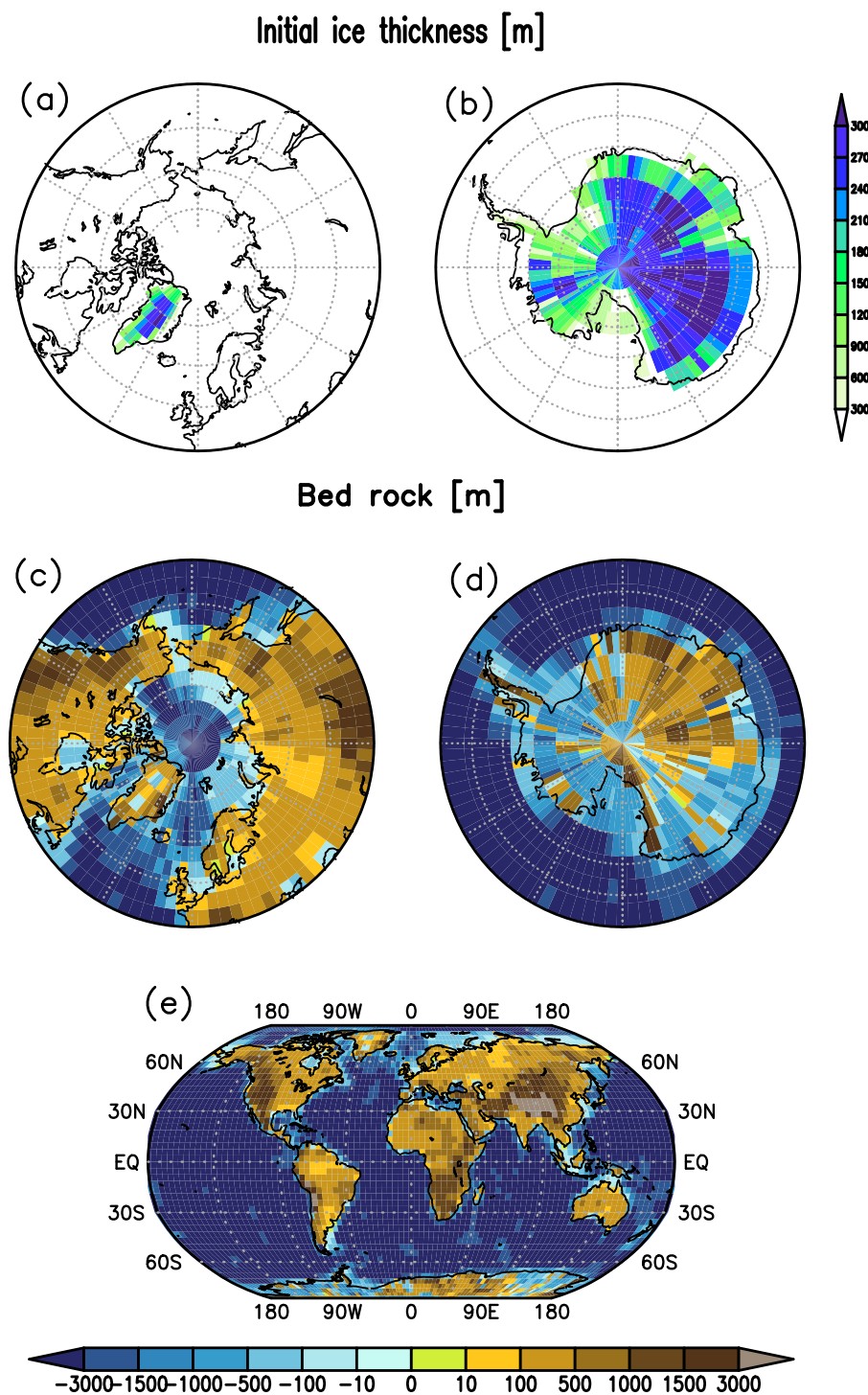

**Figure 1.** Topography in GREB-ISM.



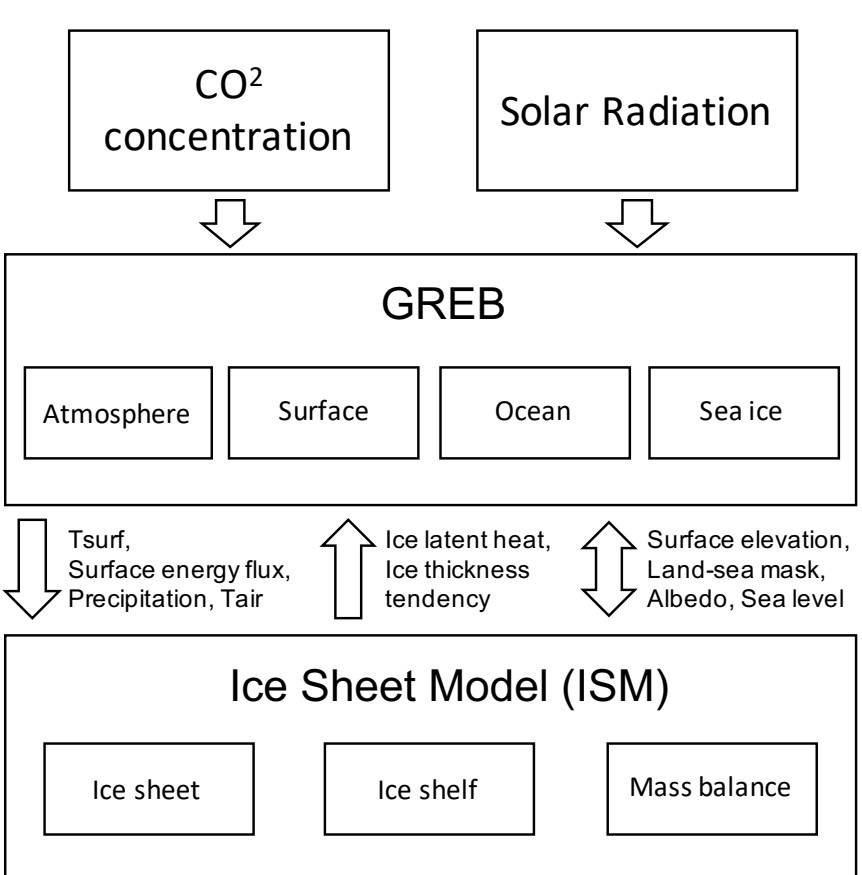

**Figure 2.** Schematic illustrating the coupled GREB-ISM.



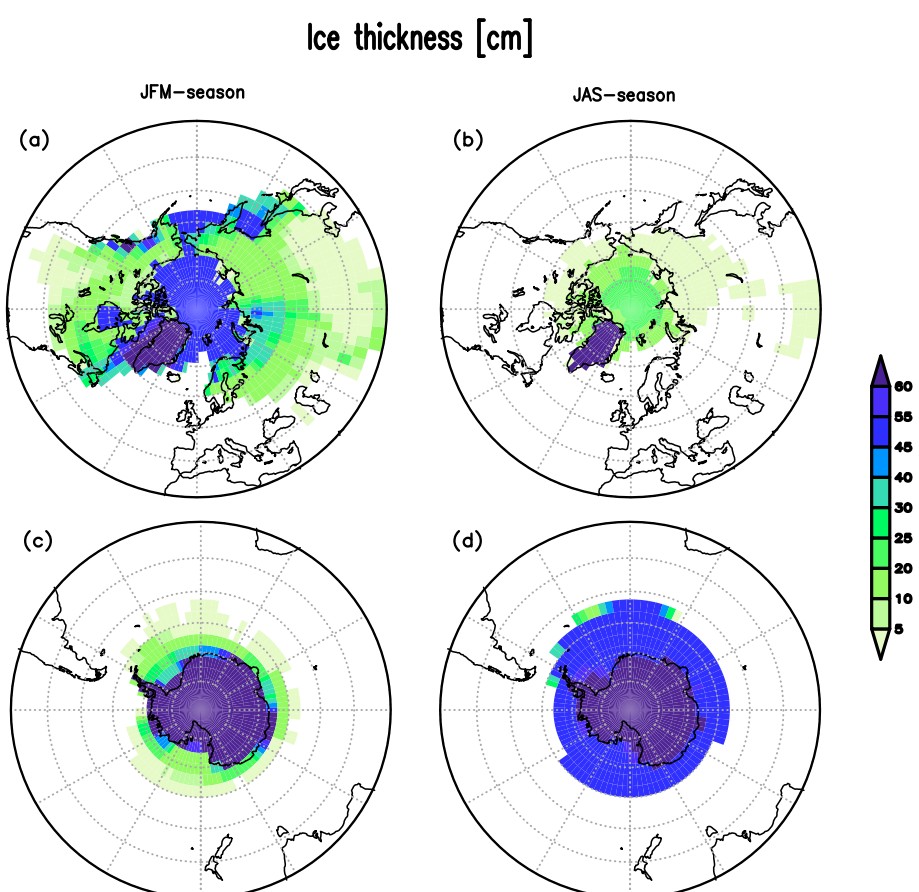

**Figure 3.** GREB-ISM seasonal ice height (units: *cm*) during January-February-March (left) and July-August-September (right).



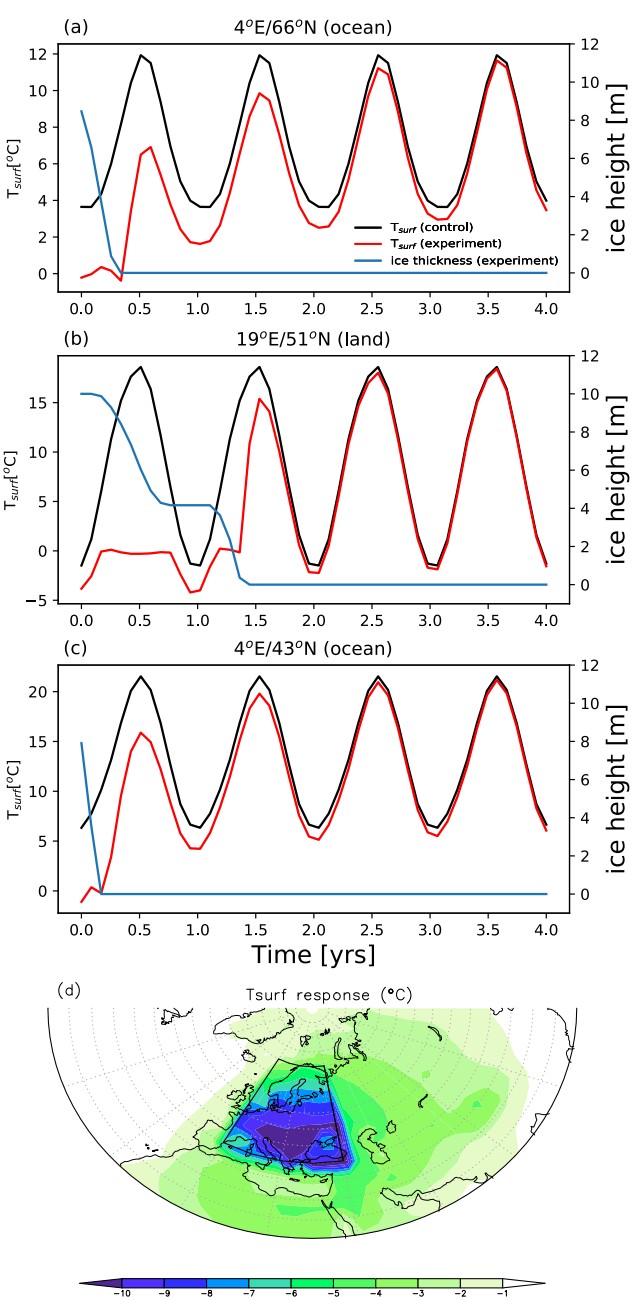

**Figure 4.** GREB-ISM surface temperature and ice thickness response to adding 10 m ice thickness experiment. (a), (b) and (c) are the temperature (units: $^{o}C$) and ice thickness (units: $m$) evolution in three different sites. (d) shows the temperature difference (units: $^{o}C$) at the end of the first simulation year after adding 10m ice thickness anomalies.





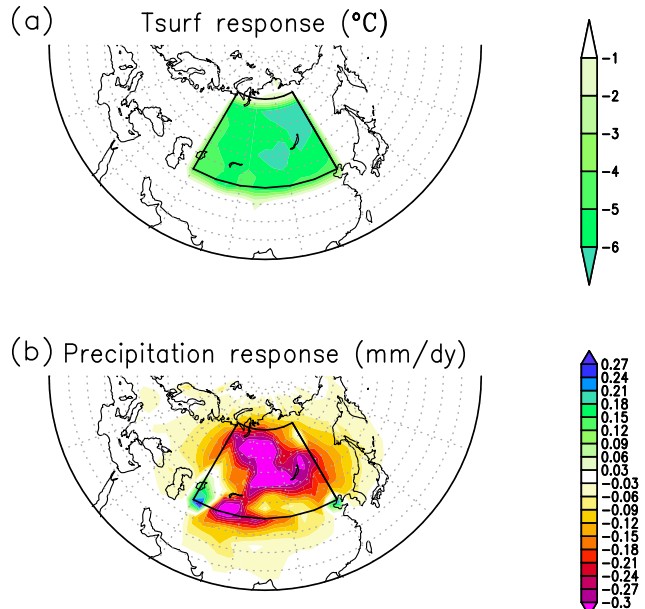

**Figure 5.** GREB-ISM surface temperature (a, units: $^oC$) and precipitation (b, units: $mm\ dy^{-1}$) response to 1000 m topography lifting experiment.



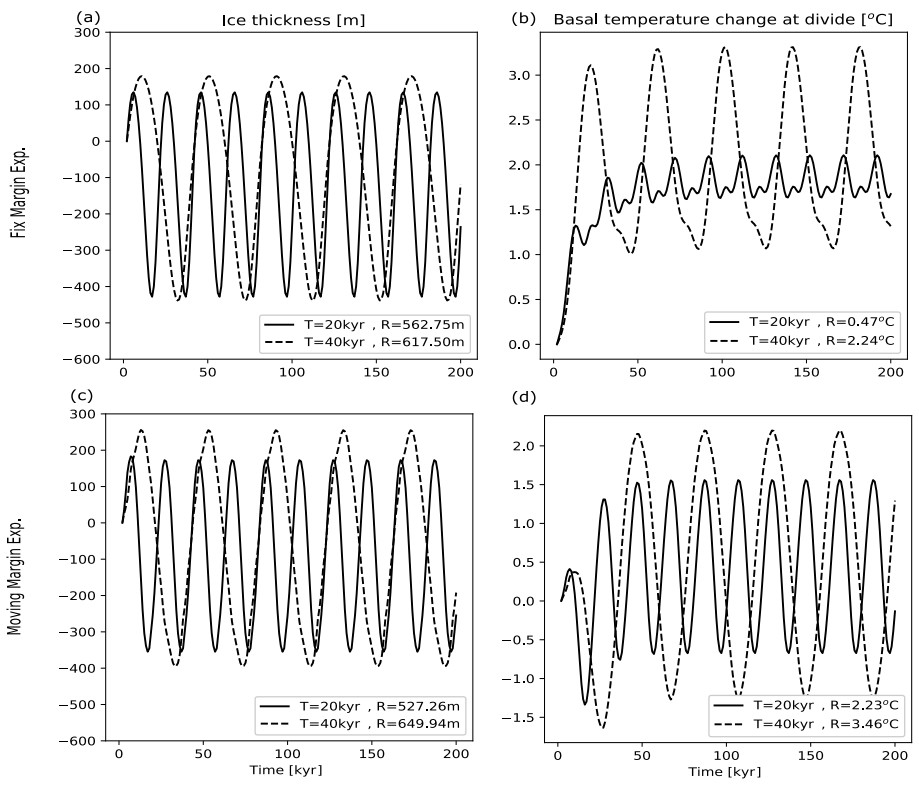

**Figure 6.** Time evolution of ice thickness (a, c, unit: $m$) and homologous basal temperature (b, d, unit: $K$) in the EISMINT I fixed (a, b) and moving margin experiments with GREB-ISM (c, d) with 20/40 kyr forcing.

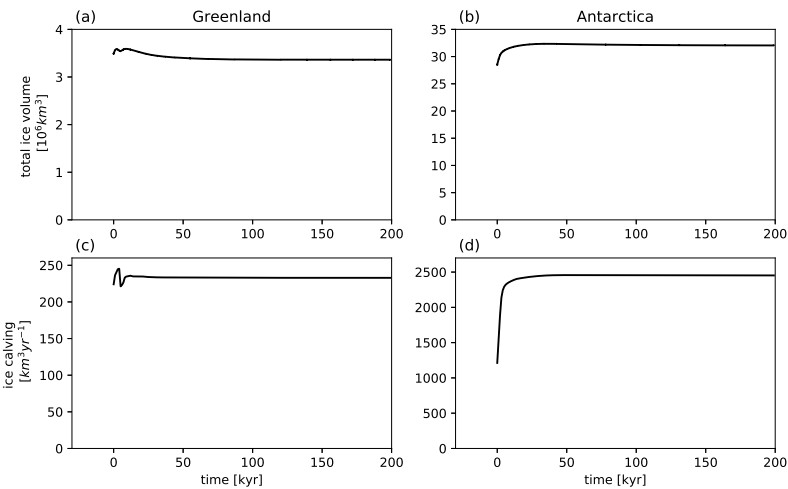

**Figure 7.** Time evolution of total ice volume (a, b, units: $10^6 \ km^3$) and ice calving (c, d, units: $km^3 \ yr^{-1}$) in Greenland (a, c) and Antarctica (b, d) from the forced stand-alone dynamic equilibrium simulation.



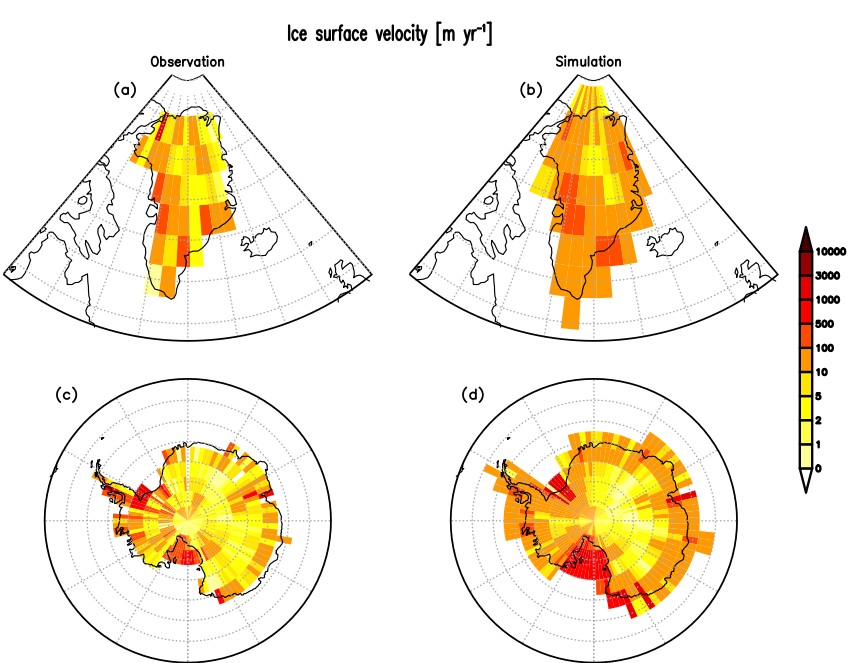

**Figure 8.** The comparison of ice surface velocity (unit: $m\ yr^{-1}$) from observations (left) and the GREB-ISM forced stand-alone dynamic equilibrium simulation (right).



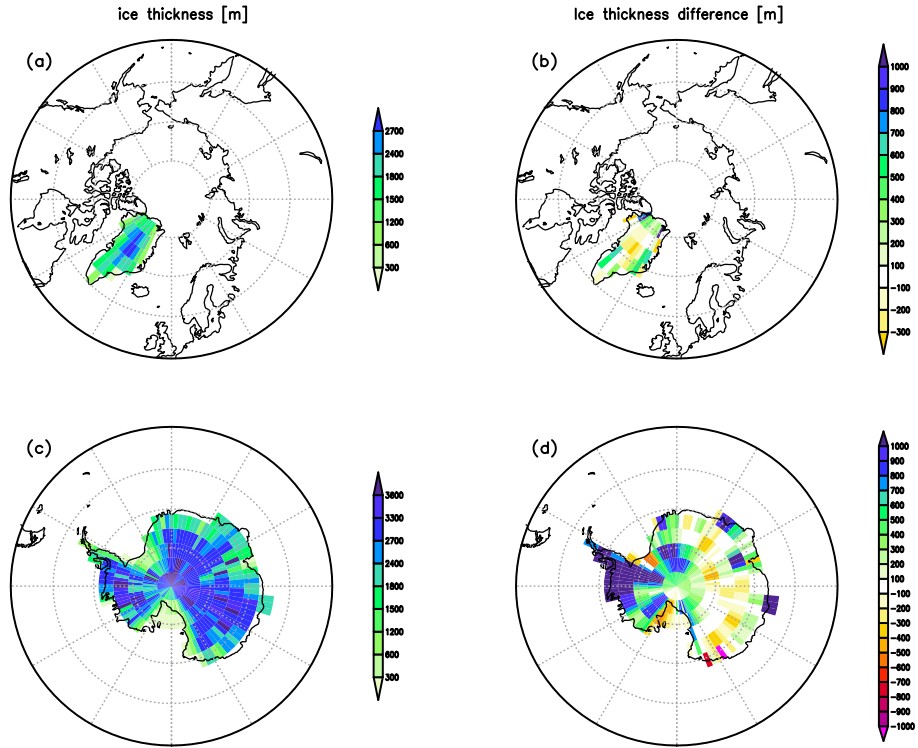

**Figure 9.** Results from the GREB-ISM forced stand-alone dynamic equilibrium simulation after 200 kyr: Annual mean ice thickness (a, c) and difference between annual mean ice thickness (b, d) and ice thickness observation derived from Bedmachine dataset in Greenland (a, b) and Antarctica (c, d).

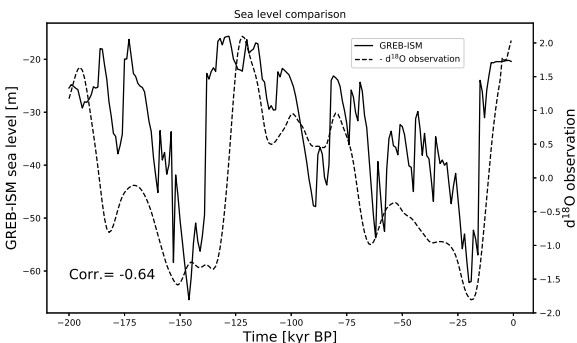

**Figure 10.** Time series of simulated sea level (left axis; units: *m*) from the GREB-ISM transition experiment and d$^{18}$O observation (right axis).





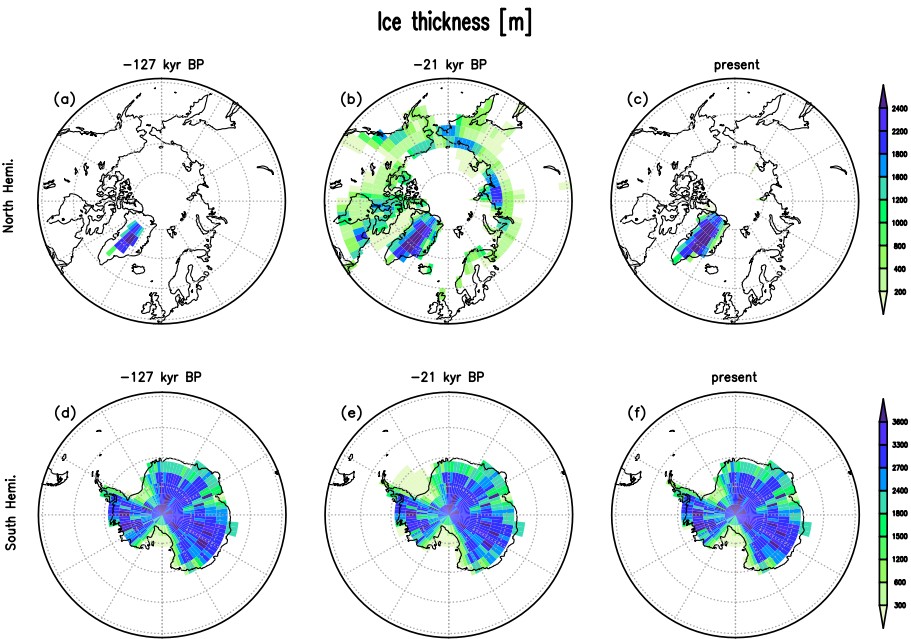

**Figure 11.** Global ice thickness (unit: $m$) distribution in the Last Interglacial (left), the Last Glacial Maximum (middle) and present day (right) from the transition experiment.

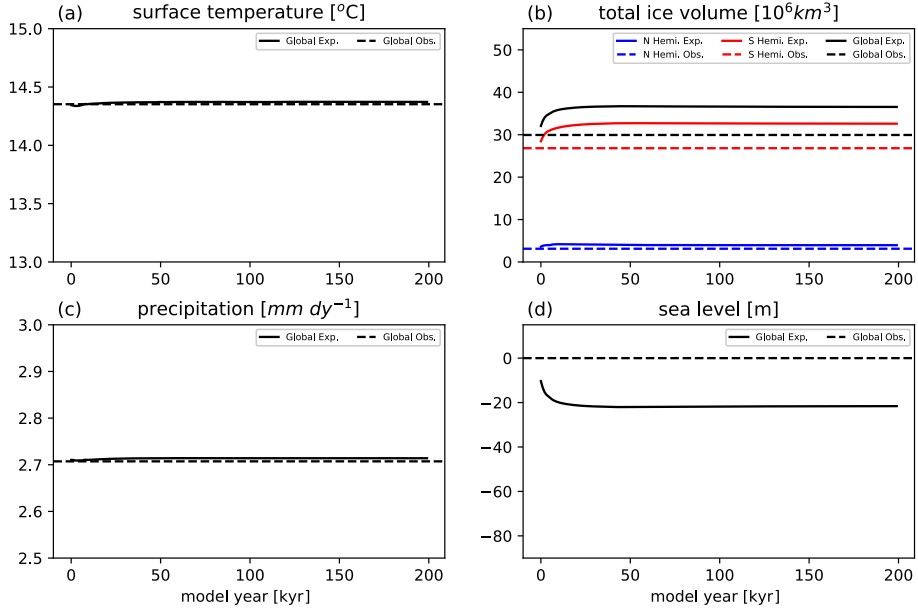

**Figure 12.** Results from the fully coupled dynamic equilibrium experiment: Evolution of global annual mean surface temperature (a, units: $^oC$), total ice volume (b, units: $10^6\ km^3$), annual mean precipitation (c, units: $mm\ dy^{-1}$) and sea level change based on current day (d, units: $m$). The dash line are modern observation references.



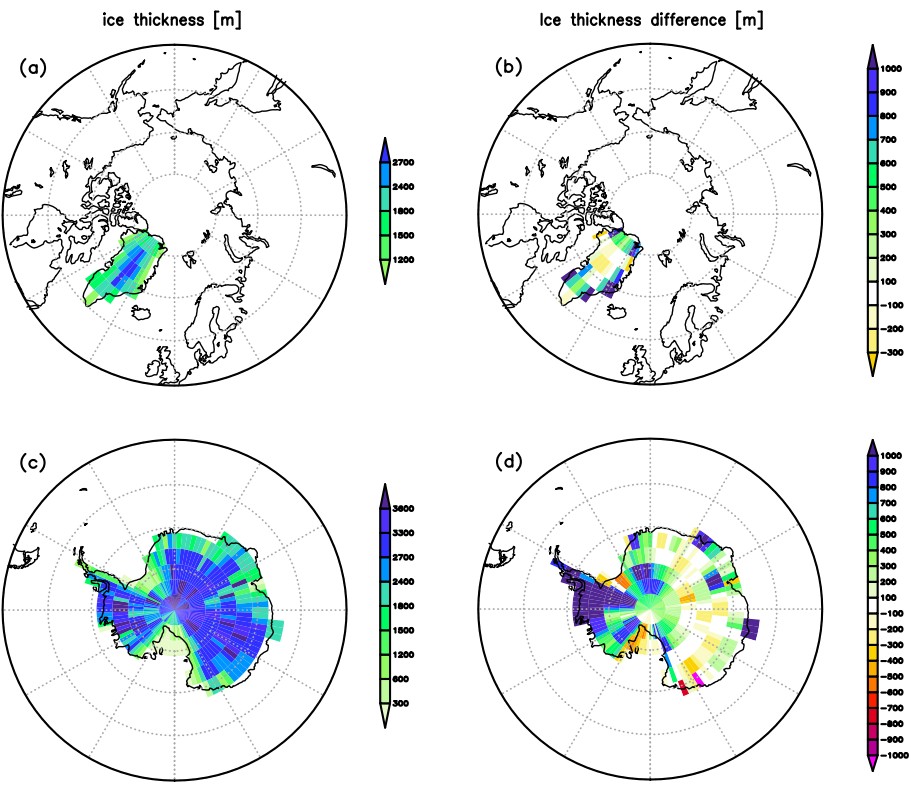

**Figure 13.** Same as Fig. 9 but for the fully coupled GREB-ISM.

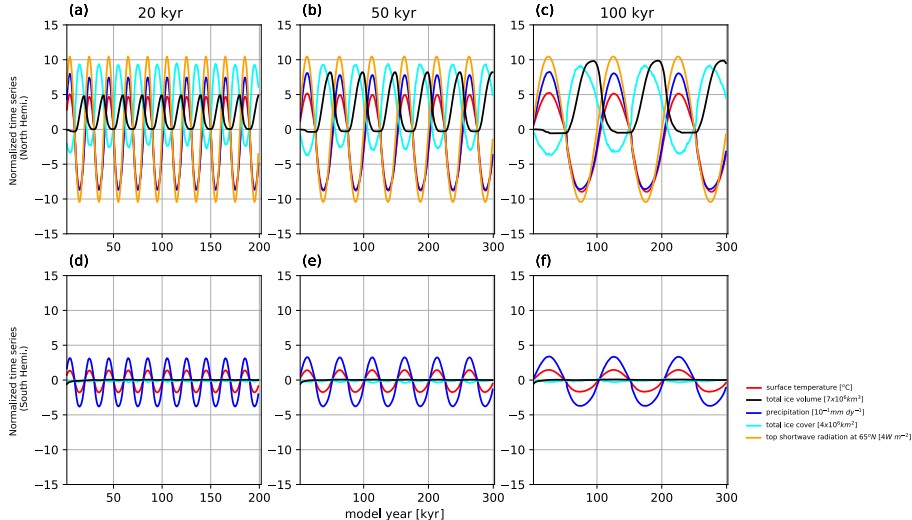

**Figure 14.** Time evolution of total ice volume (black, unit: $7 \times 10^6 km^3$), surface temperature (red, unit: $^oC$), precipitation (blue, unit: $10^{-1} mm\ dy^{-1}$), ice cover area (cyan, unit: $4 \times 10^6 km^2$) and solar radiation at 65 $^oN$ (orange, unit: $2W\ m^{-2}$) from the shortwave oscillation experiment with forcing period of 20 kyr (left), 50 kyr (middle) and 100 kyr (right).



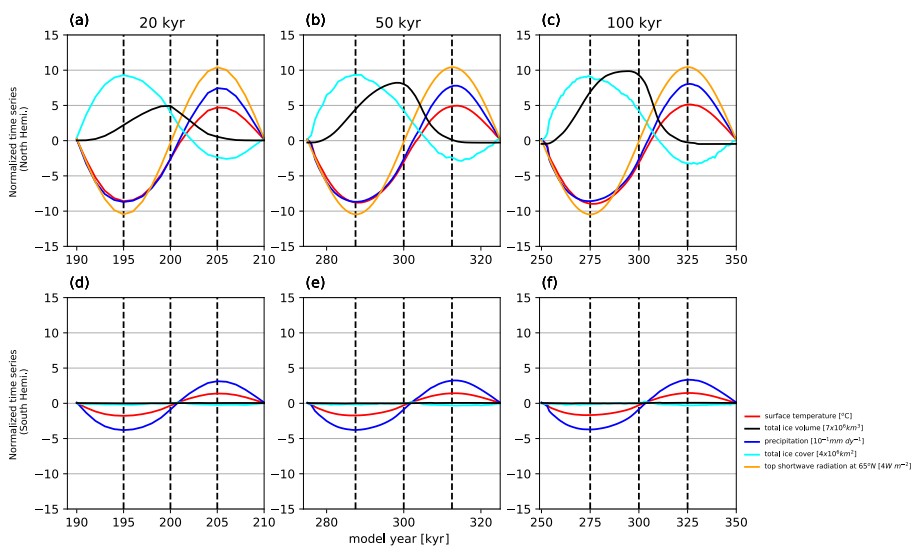

**Figure 15.** Same as Fig. 14 but at only for the last cycle of each run. The vertical dash lines represent the solar forcing sine function phases of $-90^o$, $0^o$ and $90^o$.

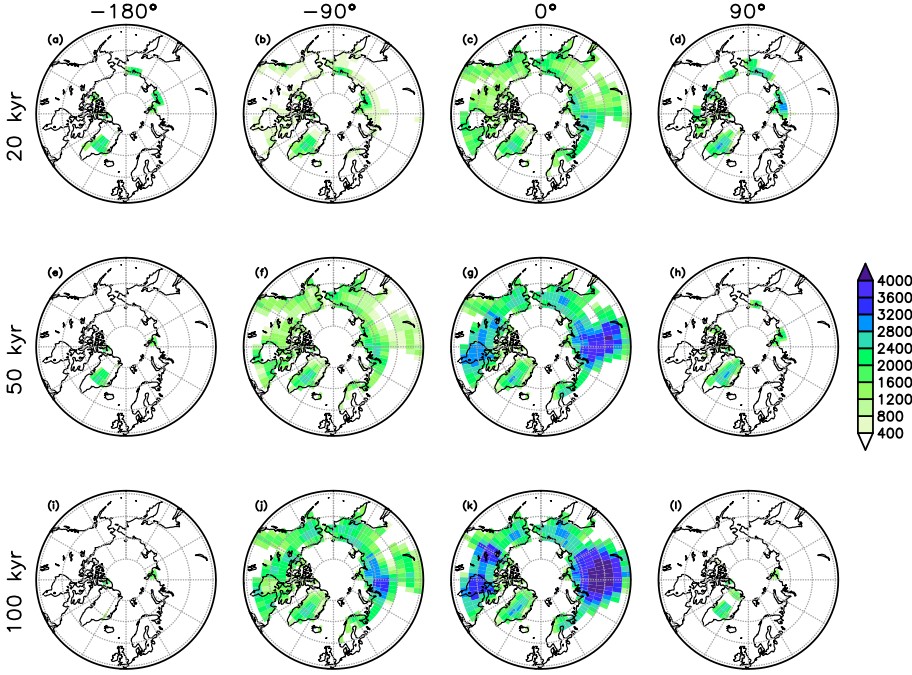

**Figure 16.** Ice thickness (unit: m) distribution in four phases within forcing period of 20 kyr (upper), 50 kyr (middle) and 100 kyr (lower) from the last cycle of the simulations. The headings mark the $-180^o$, $-90^o$, $0^o$ and $90^o$ phase of the solar forcing phases.





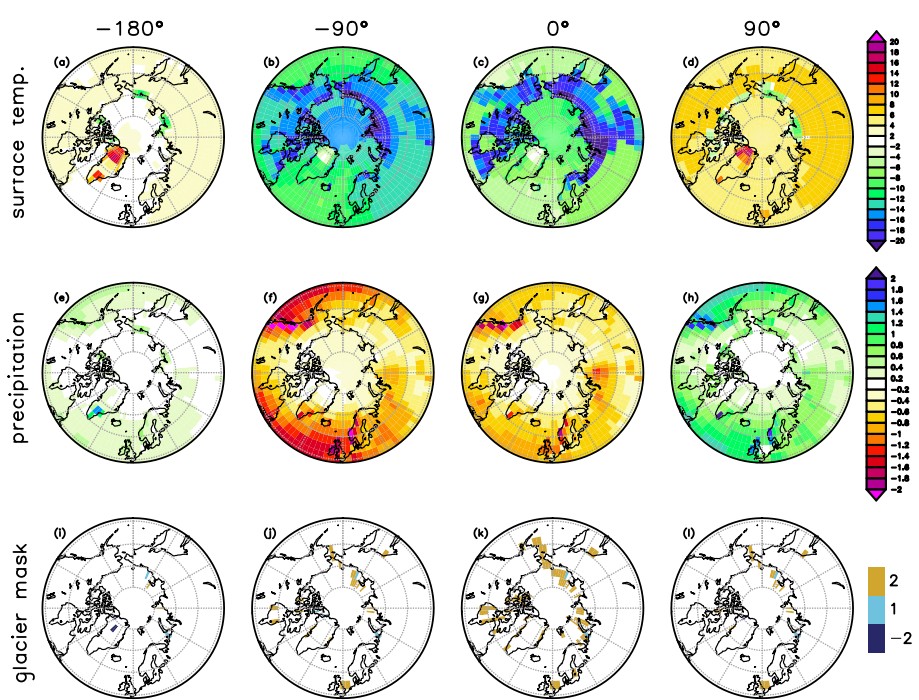

**Figure 17.** Surface temperature anomalies (upper; unit: $^oC$), precipitation anomalies (middle; unit: $mm\ dy^{-1}$) and glacier mask change (lower; 2 = from ocean to land, 1 = from ocean to floating ice, -2 = from land to ocean) in four phases during the last cycle of the 20 kyr periodic forcing experiment. The average value from the dynamic equilibrium is removed to obtain anomalies. The headings mark the $-180^o$, $-90^o$, $0^o$ and $90^o$ phase of the solar forcing phases.