# Peer review of "GREB-ISM v1.0: A coupled ice sheet model for the Global Resolved Energy Balance model for global simulations on time-scales of 100 kyr"

_Geoscientific Model Development, 2021_

## Referee Comment (RC2)

The authors introduce a newly developed global ice sheet-climate coupled model which aims to explore the evolutions and variabilities of ice sheet-climate system on very long time scale. They first describe several changes applied in the GREB climate model and then introduce a new ice sheet model. Several suits of test experiments are performed to assess the ability of the model in simulating responses of ice sheet-climate system to idealistic forcing. They conclude that the model reproduces the general behavior of ice sheet variabilities reasonably well and state that it can be used for studies on very long time-scale, such as ice sheet-climate evolution over the Quaternary.

The manuscript is well written, easy to follow and its content matches the concept of GMD. In addition, as a paleoclimate modeller using comprehensive AOGCMs, I find this study and model quite interesting since it enables the community to explore the interactions and variabilities of ice sheet and climate in a global scale. On the other hand, I also find several concerns in the model as well as ambiguities in the aim of sensitivity experiments. For these reasons, I recommend major revision before accepting the paper for publication. Below summaries my main concerns, followed by specific comments.

**Major comments**
1. My main concern appears in the ambiguity on what kind of insights could the model provide in future studies. This model is clearly not meant to reproduce the ice sheet and climate precisely, but rather meant to reproduce them crudely so that the authors can discuss their interactions on global and long time scale. However, there is still a big problem in the spatial distribution of the simulated ice sheet; thickest glacial ice sheet exists over Siberia (Figs. 10 and 16). Previous studies have shown that the longitudinal locations of ice sheets affect their dynamical characteristics, which is important for interpreting the evolution of glacial-interglacial cycle (e.g. Abe-Ouchi et al. 2013, nature, Fig.2b). Given that, how reliable a model with such a bias could be in discussing dynamics and evolutions of ice sheet and climate (e.g. glacial-interglacial cycle across the MPT, the effects of regolith on the ice sheet and so on).Therefore, I strongly encourage the authors to 1. put more efforts to reduce the thickest ice sheet in Siberia and 2. to discuss on what kind of insight could the model provide in understanding the ice sheet-climate system of the Quaternary.

2. While the ice sheet model simulates the ice shelf component of the ice sheet, I couldn't find discussions on the effect of basal melting at ice shelf-ocean margin. Perhaps this effect is partly incorporated through sea ice dynamics, but the authors should discuss possible effects of not including ice shelf basal melting in the model, especially over Antarctica. I assume this is partly causing the too thick west Antarctica in modern climate. Also, it might be related to the small sensitivity of Antarctic ice sheet to climate changes (L466-467).

3. The aim of transient test experiments of ice sheet is unclear to me (L434-L437). The authors apply a reconstructed Greenland temperature used in Greve (1997) to their ice sheet model over the entire northern high latitudes. This method may be appropriate for Greve (1997), which focused on Greenland ice sheet, or a 2D (latitude and height) ice sheet model. However, it is clearly not suitable for a study focusing on 3D Northern Hemisphere ice sheets, since it overestimates/underestimates glacial-interglacial temperature differences outside the Greenland area (e.g. Kageyama et al. 2021, Climate of the Past, Fig. 2b, https://doi.org/10.5194/cp-17-1065-2021). Partly as a result, the thickest ice sheet develops over Siberia, which is affecting the simulated

sea-level change. I would recommend the authors to use different methods such as that used in Niu et al. (2019, Journal of Glaciology, doi: 10.1017/jog.2019.42). In this study, they force the ice sheet model with a similar index method, but utilise information from climate models in considering the regional dependences in temperature changes. In this method, one can assess whether the ice sheet model can simulate the geographical location and volume of ice correctly when reasonable atmospheric forcing is applied.

4. Metrics of assessing model's reproducibility are sometimes vague to me. For example, the authors state their model generally reproduces the ice velocity (Fig. 8 & L432) and sea level change reasonably well, but I couldn't find how they defined that (sorry if I missed). Is it because the model reproduces the order of magnitude correctly? Please add an explanation on this.

**Specific comments**
L30: "procession" -> "precession"?

L53: Perhaps you may also cite a recent study by Willeit et al. (2019, Science advances, eaav7337) here.

L91 and Table 1: I don't understand what "air thermal heat" means. Could you add an explanation in the main text or table?

L99: It may be useful to add a sentence describing how you estimated the correction terms for temperatures. (Sorry if I missed them)

L140: Is there any reason behind the choice of 10 m threshold? If so, please add a short explanation.

L168: What is Lice? Is it explained in the manuscript or is it a typo of Lm?

L180 and Fig.3: How does the simulated ice cover (or thickness) compare to modern observation? This tells the biases in the model, which is important to know when interpreting the evolution of the ice-climate system of the past.

L272-273: Making a bold assumption is sometimes inevitable in relatively simple models, but the authors should present some evidences supporting this assumption.

L288 and equation (32): Why do you set a maximum ice thickness for ice growth? Please add a sentence on it.

L379: Would be helpful for readers if you refer to Table 3 here.

L398: Do you mean Experiment B is warmer by 5K compared to Experiment A?

L423: Isn't it simply because your model does not account for strong ice shelf basal melting in West Antarctica? Perhaps you can crudely account for this effect by increasing the calving in this area.

L447: As far as I know, Greenland ice core data extends back to 130ka. How did you force the ice sheet model before this period. Perhaps, it may be written in Greve (1997), but please add a sentence on this point.

L453: Do you have any possible explanation why the model underestimate the amplitude of the sea level variability? A sentence would be sufficient.

L459: "EIVM", do you mean by LIG here?

L465: Do you mean "our simulation of"?

L516: Why does the mean ice volume increase after the first cycle?

L564: Could you further elaborate on how the coarse resolution of ice sheet model causes the bias in West Antarctica?

L567-568: Since the authors use Greenland temperature record to force the ice sheet model, I assume that the effect of changes in ocean circulation on ice sheets are already partly incorporated. Could you further elaborate on how the absence of ocean circulation affects the the biases in simulated North American and Eurasian ice sheet?

---

## Author Response (AR1)

**Revisions of "GREB-ISM v0.3: A coupled ice sheet model for the Global Resolved Energy Balance model for global simulations on time-scales of 100 kyr"**

Dear Editor and referees,

We would like to thank the two referees and the editor for their time spent on reviewing this manuscript, and for the many very helpful comments they provided. We think the referee comments have helped us to substantially improve the presentation of this work.

For the new manuscript and model, we adopted the suggestion from Referee #2 and changed our standalone transition experiment boundary condition, which has greatly improved this sensitivity experiment. Meanwhile, we fixed several minor bugs in the model code that did have a minor effect on the coupled model simulations. Below we give a point-to-point response to all referee comments, hoping the revised manuscript has now been improved in clarity and is ready for publication.

With best regards,

Zhiang Xie, Dietmar Dommenget, Felicity S. McCormack, and Andrew N. Mackintosh

*This paper presents the new capability of the simple GREB model being coupled to an ice sheet model (ISM). This presentation introduces the new ISM itself and its coupling to GREB. The authors use a series of stand-alone ice sheet model intercomparison projects (EISMINT) to test their new ISM and a series of simple coupled simulations to investigate the feedback between the ISM and the climate system. This model has the great advantage to be computationally very cheap.*

*The paper is written in a logical fashion and is well organized. I will begin with general comments before addressing specific minor details.*

**Response:** We like to thank the referee for the evaluation of our manuscript and the many comments that helped us to improve the manuscript.
* * *
*general comments:*

*Overall, I thought the authors were trying to combine too many model development details in a relatively short manuscript. While I appreciate the will to keep the paper concise it results in a lack of convincing arguments to make the reader believe the ice sheet model is behaving in a sensible way especially since one the goal here is clearly to capture sea level change.*

**Response:** We do acknowledge that our presentation was in many parts too short. We have now substantially reworked the manuscript and extended the discussions to better illustrate the arguments for some of the modelling strategies and our sensitivity experiments. While sea level simulation is an element of the GERB-ISM, we would not consider this the main goal of the model. We primarily aim at simulation of past, Quaternary period climate variability.
* * *
*In particular, the paper lacks in basics sensitivity experiments that could shed more lights on some model limitations that are inferred in the text. For example, in the EISMINT experiments,*

*the authors hint that the coarse vertical resolution might impact the strong differences in basal temperature. The model being very cheap, I would appreciate seeing a sensitivity to rule this out.*

**Response:** Please note, that the aim of this study is to provide a model tool to understand ice-age cycle on 100kyrs time-scale, for the global climate and the corresponding physical mechanisms. Our study focusses on illustrating the fully coupled model for this aim. The EISMINT experiments are one element in testing this, but given they only focus on a higher-resolution problem for only a small region they are not the most important ones. We therefore have to strike a balance and have to focus more on the global coupled system for which we have presented a number of different sensitivity studies.

Nevertheless, we tried to improve the discussion of the EISMINT experiments. In order to verify our assumptions that the basal temperature bias comes from vertical resolution, we rerun the EISMINT experiments with 10 layers. For EISMINT I (see Table R1), the basal temperature with 10 layers from ISM is much closer to the results in H96 than that with 4 layers. As for EISMINT II (Table R2), increase of vertical resolution leads to a higher percentage melt fraction, closer to results from P2000. These results provide good support for our assumptions.

We further also did some test with the horizontal resolution, which does improve the mass flux at the midpoint. We have now mention these results in the manuscript to give the reader some idea of what is causing the GREB-ISM model limitations in the EISMINT experiments.

**Table R1.** EISMINT I result comparison between GREB-ISM and the model ensemble from H96. F and M in the experiment description represent fixed-margin and moving-margin experiments, respectively. -4 for 4 layers result, -10 for 10 layers.

| Experiment | ice thickness at divide m | Mass flux at midpoint $10^2$ m$^2$a$^{-1}$ | Basal temperature at divide °C |
|---|---|---|---|
| GREB-ISM (F-10) | 3399.06 | 750.14 | -8.98 |
| GREB-ISM (F- 4) | 3399.06 | 750.14 | -11.74 |
| EISMINT I (F) | $3384.4 \pm 39.4$ | $794.99 \pm 5.67$ | $-8.97 \pm 0.71$ |
| GREB-ISM (M-10) | 2916.025 | 1234.40 | -12.75 |
| GREB-ISM (M- 4) | 2916.025 | 1234.40 | -14.93 |
| EISMINT I (M) | $2978.0 \pm 19.3$ | $999.38 \pm 23.55$ | $-13.34 \pm 0.56$ |

**Table R2.** Results for basic glaciological quantities in EISMINT II experiments after 200 kyr. Differences are defined as current experiment minus experiment A. Percentage changes are relative to experiment A. The results of P2000 are shown in the form of "mean ± range". See text for details. 4 inside the brackets for 4 layers result, 10 for 10 layers.

| Model (Exp. label) | volume $10^6$ km$^3$ | area $10^6$ km$^3$ | Melt fraction | Divide thickness m | Divide basal temperature K |
|---|---|---|---|---|---|
| GREB-ISM(A10) | 1.9997 | 0.932223 | 0.53031 | 3723.49 | 255.343 |
| GREB-ISM(A04) | 2.065 | 0.932 | 0.466 | 3829.77 | 254.038 |
| P2000(A) | 2.128 ±0.145 | 1.034 ±0.086 | 0.719 ±0.290 | 3688.342 ±96.740 | 255.605 ±2.929 |
| Model (Exp. label) | volume change % | area change % | Melt fraction change % | Divide thickness change % | Divide basal temperature K |
| GREB-ISM (B10) | -3.72541 | / | 31.7021 | -5.62864 | 4.53702 |
| GREB-ISM (B04) | -4.066 | / | 38.642 | -5.821 | 4.576 |
| P2000 (B) | -2.589 ±1.002 | / | 11.836 ±18.669 | -4.927 ±1.316 | 4.623 ±0.518 |
| GREB-ISM (C10) | -25.9751 | -17.0786 | -66.3961 | -12.244 | 3.79523 |
| GREB-ISM (C04) | -25.907 | -17.079 | -100 | -12.137 | 3.856 |
| P2000 (C) | -28.505 ±1.204 | -19.515 ±3.554 | -27.806 ±31.371 | -12.928 ±1.501 | 3.707 ±0.615 |

 *Additionally, the horizontal resolution of the ISM is increased in the EISMINT experiments to better compare the results to the one published (I assume). Typically, good results at coarse resolution translates to as good or better results at higher resolutions. However, you plan on using your ISM on a much coarser global grid than the one used in the EISMINT experiments. I would have liked to see a comparison at that resolution as well.*

**Response:** The reviewer is right, we changed the GREB-ISM grid to match with the EISMINT experiment setting. If we use GREB resolution setting, we have only two grid cells in the EISMINT domain, which limits the extent to which a useful comparison can be made between the GREB-ISM and EISMINT simulations. The EISMINT experiments have only a limited scope and application, which is why we also present a series of other sensitivity experiments. Our dynamic equilibrium and paleoclimate transition experiments are designed to show the model capability to reproduce realistic ice sheets and related dynamics (transports) in the GREB-ISM grid resolution and parametrization. It turns out that our ice thickness simulation matches well with the observation in Greenland and most of Antarctica (e.g., Figs. 8, 9 and 13).

We have improved the discussion of the EISMINT experiments and also the other experiments to illustrate the skill of the new GREB-ISM in section 4.1 and 4.2. Ultimately, the GREB -ISM

real skill can only be evaluated if it is applied to real research problems, while here in the introduction paper we can only give some indication.
* * *
*There is a lack of discussion about the consequences of using such a coarse horizontal resolution for ice sheet modeling while the literature has shown abundantly that a grid resolution of at least 4km if not higher is necessary to simulate ice sheets (Pattyn et al. (2012), Pattyn et al. (2013),  Gladstone et al. (2010), Leguy et al. (2014), Seroussi et al. (2015), Seroussi et al. (2018), Cornford et al. (2020), Leguy et al. (2021), …). This is especially important to simulate marine ice sheets like the West Antarctic Ice Sheet for which GREB-ISM show large differences compared to observation.*

**Response:** The reviewer points out a number of studies that aim at regional scales like hundreds of kilometers. The reviewer also points at "*simulate marine ice sheets like the West Antarctic Ice Sheet*". All of these are important aspect of ice sheet modelling, but they are not the primary aim of the GREB-ISM, as we have mentioned in the title, the first sentence of the abstract, the introduction and the summary section: We aim at simulating the time evolution of global-scale ice sheets on time scales of 100 kyr. For these kinds of problem, one has to make compromises to achieve feasible speed on the computing system. Past studies that addressed similar kinds of time scales to the ones we are interested in, also use coarser resolution models (Abe-Ouchi et al., 2007; Ganopolski et al., 2010; Willeit and Ganopolski, 2018).  And as a result, the high resolution such as 4 km is not yet feasible on these timescales (Fyke et al., 2011).

We do provide some discussion on the coarse resolution in respect to the simulation of West Antarctica and in more general. We hope the revised text does better highlight this. Please see Sections 4.3 and 6.
* * *
*Many aspects of the ISM and choices are not clearly stated. For example, you choose to use the basal sliding velocity from Greve 1997. I am not saying you should not, I am simply asking to justify your choice while so many sliding laws are now available and studied (Weertman (1957), Schoof (2005), Schoof (2007), Aschwanden et al. (2013), Leguy et al. (2014, 2021), Tsai (2015), …).*

**Response:** We have improved the overall discussion of the model development and the sensitivity experiments (see section 4.3). We hope the revised version give a more complete presentation of the model. The sliding law from Greve (1997) is one of Weertman type power law (Weertman, 1957).This type of explicit form sliding law is efficient and widely used in paleoclimate research (Abe-Ouchi et al., 2007; Fyke et al., 2011). In respect to the basal sliding law we added some discussion in Section 4.3 in which we argue that the variations of the basal sliding parameters have no significant impact on the model simulations in the context of our applications.

[Figure]

**Figure R1.** Ice thickness difference [m] between models within different sliding strength and the GREB-ISM current version ($C_{sl} = 6.0 \times 10^4$) in forced dynamic equilibrium experiments. $C_{sl}$ on the title represents the sliding law coefficient value [$a^{-1}$]

To test the model sensitivity to sliding law, we run dynamic equilibrium experiments by varying sliding law coefficient (change Csl in Eq. (22)) from $6 \times 10^3$ to $6 \times 10^5$. Compared with GREB-ISM simulation, only when Csl is larger than $3 \times 10^5$ a$^{-1}$, we can see a significant ice

thickness reduction on Antarctica (Fig R1). Increasing sliding coefficient does help reduce the WAIS extra ice growth, but the East Antarctic Ice Sheet loses too much ice at the same time, which worsens slight negative bias in East Antarctica (Fig 13). So our current sliding law coefficient is still a reasonable choice. Hence, our experiments here suggest that stronger ice sheet sliding is unlikely a solution for WAIS positive ice thickness bias.
* * *
*…Also, you mention in many places in your paper how well simulated calving matches observation, while you never clearly define how you actually calve the ice (unless I missed it in which case I apologize). In section 3.2.2, you mention that a condition for floating ice shelves is that H>=10m. Does it mean that 10m is a thickness threshold below which ice calves?*

**Response:** We had previously defined calving at the end of Section 3.2.3 (Mass balance). To make this clearer, we now define a subsection 3.2.3.1 (Calving) to better highlight how we diagnose calving.
* * *
*There is also no discussions of any sub-grid scale parameterizations (for grounding lines, calving fronts, …), does the reader need to understand they aren't any?*

*I do note that the authors left a more in-depth study of the ISM properties for another publication, but if so, I almost feel like it should have come before publishing this paper to rule out many possibilities leading to large differences in the results.*

**Response:** If we have specific sub-grid parameterization scheme in our model, we will explicitly explain it in the text, such as snowfall rate and sliding law. So, we do not include those sub-grid parameterizations you mentioned yet. The grounding line in our model is decided by the glacier type mask, which is defined in section 3.2.2. We have now added a sentence (section 3.2.2) to explicitly point out that grounding lines of glaciers are defined by shifting points from grounded ice to floating ice and therefore change the method of how the dynamics of the ice flow is simulated. For the simulation of calving, please see our response to the previous comment.

We have also tried to improve our presentation in many other aspects to better illustrate how

we model the ice and how the different sensitivity experiments illustrate the skill and limitations of the model.
* * *
*I would also have appreciated a discussion on the strength and weaknesses of the model and emphasize a bit more the type of experiments for which the model is worth "trusting", and those that is best not. For instance, Figure 10. Clearly shows that the model is good at capturing trends and in some cases with a time lag of several 10th of thousands of years. This indicates that the model is not best used for short time periods whatsoever.*

**Response:** We tried to improve the discussion of limitations and applications of this model throughout the manuscript. We hope the revised text does give a better presentation.
* * *
*Finally, I would encourage the authors to improve their figures. I thought the choice of the color scale to be difficult to distinguish, especially the green scale and some labels hard to read. The figure captions need to contain more details for some of them. I will add more details for specific figures below.*

**Response:** We worked through all the figures again and improved the presentations. See also comments below.
* * *
*Minor Comments:*

*P2, l48: "Ice sheet modelling …". It really depends on what you are doing, and some models are capable in simulating changes on century time scales. Please rephrase.*

**Response:** We rephrased this section to better highlight the long and large-scale ice sheets simulations we are interested in.
* * *
*P2, l59: replace CO² with CO₂.*

**Response:** Corrected.
* * *
*P2, l60: add "(ISM)" after "ice sheet model" as you will use it later (Fig2) and never defined the acronym. It is best to define in the text as opposed to in a table.*

**Response:** This is now included.
* * *
*P4, l11: "state-independent". I think this deserve a longer explanation and clarification as it is key to understand your coupled experiments.*

**Response:** We added a sentence in section 3.1 to explain a bit more the concept of state-independent flux corrections. We also added some references, as this is a widely used concept.
* * *
*P4, l103: spell out CGCM.*

**Response:** Done!
* * *
*P4, l115: remove and reword "due to limited space". GMD is not page limited.*

**Response:** We reworded this sentence.
* * *
*P4, l124: "Four vertical layers are chosen". Why only 4? See general comment above.*

**Response:** We have now added a short statement on why we chose 4 layers. We also discuss this in the EISMINT experiments. 4 layers are selected as it is close to minimal number of layers that can still resolve the vertical velocity in the ice sheet.
* * *
*P5, l140: What happens when H<10m?*

**Response:** If $10 \, \text{m} \geq H \geq 0$, we assume that ice sheet flow is negligible. We have added a few sentences to better explain this.

*P5, l141: Add below that H is used for both sea ice and ice sheet. You mention this way later.*

**Response:** Done!
* * *
*P6, l156: space out "Tm" and "and" in the equation.*

**Response:** Done!
* * *
*P6, equation 11: what happens when Fsurf=Fmaxmelt?*

**Response:** The situation is included in *Fsurf<Fmaxmelt* case. The condition for first equation is set as $F_{surf} \leq Fmax_{melt}$.
* * *
*P6, l176: how do you convert snow to ice?*

**Response:** Over land and ice sheet grid cells, equation 8 states how snow is converted to ice. This is then coupled with equation 7 that shows how land ice thickness changes with time. For sea ice, snow is ignored, as it is a smaller forcing than the other terms in equation 31.
* * *
*P6, l180: This is a bit confusing. In page 4, line 115 you refer to table 2 which describes the time step of the ISM which is 1 year. How do you obtain a seasonal cycle? Please, better describe what you are doing here.*

**Response:** The surface mass balance equation 7 is solved with a half-day time step, as described in Table 2. So due to ice accumulation and melting, we have a seasonal cycle of thickness change. For ice sheet dynamic part (i.e. the mechanical model described in Section 3.2.4), the integration is in 1-year time step. We now explicitly point this out in Section 3.2.3 (Mass balance).

*P7, l186: This sentence is confusing, please reword. You do not show the Full Stokes equations here.*

**Response:** We simplify the sentence and hopefully it is clearer now (section 3.2.3).
* * *
*P7, equations 16 and 17: add a reference to which axis each equation needs to be solved for.*

**Response:** The equation is solved in geo-coordinate, which is now added in the text.
* * *
*P7, l202: Correct " T' " to match the one in equation 19.*

**Response:** Corrected.
* * *
*P8, l212: Can you motivate your choice of sliding law? (See comment above)*

**Response:** The sliding law from Greve (1997) is one of Weertman type power law (Weertman, 1957). This type of explicit form sliding law is efficient and widely used in paleoclimate research (Abe-Ouchi et al., 2007; Fyke et al., 2011). See response above.
* * *
*P8, l232: Can you motivate your choice of constant geothermal heat flux while observation is available?*

**Response:** We have now added a short discussion on why we have chosen a global constant. Previous studies show that the model with uniform geothermal heat flux is still able to reproduce the ice sheet evolution in the paleoclimate (Abe-Ouchi et al., 2007; Tigchelaar et al., 2019). This is mostly done to be consistent for all global ice sheets, and the varying geothermal heat flux can be set in the future.
* * *
*P9, l247: how long will you run this simulation?*

**Response:** The simulation is totally 4 years. We mention this now in the text.

*P10, l266: please add this equation here to limit the number of papers the reader has to look at to understand yours.*

**Response:** The original equation has been added as eq. (30) and some additional explanation about the precipitation model has been added.
* * *
*P10, eq.32. what happens to the energy that would typically be used to grow the sea ice but can't because of the 0.5m threshold?*

**Response:** If the sea ice is more than 0.5m, the energy will be used to cool down the surface temperature instead of forming more sea ice. We have now added a bit more explanation in the text to explicitly point this out.
* * *
*P12, l340: can you justify this value of 0.3?*

**Response:** This is the mean observed value from the climatology used in Dommenget and Floeter 2011. This is now stated in the text.
* * *
*P13, l363: I don't understand the justification of EISMINT being designed on a cartesian grid in order for you to regrid your ISM. Please rephrase.*

**Response:** Our text was a bit unclear here. We did use a spherical geocoordinate grid. We only changed the meridional resolution to match those of the EISMINT simulations. We have revised the text to better describe this.
* * *
*P13, eq. 37 and 38: please define the parameters in a table.*

**Response:** Done. You can find the parameters in Table 3.
* * *
*P13, l376: can you comment on the mass flux being 20% higher than EISMINT results?*

**Response:** We have now mentioned some additional experiments in the manuscript in section 4.1, which illustrate that this mass flux value is depending on the horizontal grid. If we chose

a Cartesian gid similar to those of the EISMINT simulations, then the mass flux in the GREB-ISM is the same as in H96.
* * *
*P13, l378: replace "from" with "for".*

**Response:** It has been corrected.
* * *
*P13, l380: see comment above.*

**Response:** It has been discussed above.
* * *
*P14, l404: delete "roughly".*

**Response:** Done.
* * *
*P14, sec4.3: What do you invert for in these spin-up experiments? You have not mentioned anything about Csl at this point. If you invert for this field, please provide a figure. What years does your climatology span?*

**Response:** We use the *Csl* as in Greve (1997). We did some sensitivity experiments to see if variation in *Csl* could improve the overall simulations, in particular over West Antarctica, but no improvement could be found. We now mentioned this in section 4.3.
* * *
*P15, l423: See comments above. Grid resolution, choice of sliding law, … could be part of the cause of your limitation.*

**Response:** We have revised some of our discussion in the text and hope this is now better presented. See also our response to the comments above.
* * *
*P15, l434: You can omit this first sentence and begin your paragraph directly with "We next evaluate the capability of the global…"*

**Response:** Your suggestion has been adopted.
* * *
*P16, paragraph 2: big caveat to simulate sea level change. See comment above.*

**Response:** We have changed the set up of the transition experiment and the current sea level variations are much more realistic.
* * *
*P16, l459: spell out EIVM.*

**Response:** EIVM is replaced by Last Interglacial (LIG).
* * *
*P16, l465: add "of" after simulation.*

**Response:** Corrected.
* * *
*P16, l469: "It is beyond …" I don't think it is beyond this study to explore these deviations, since your model is very cheap to run and you're trying to validate the use of your ice sheet model configuration for future studies. See comment above.*

**Response:** We revise this simulation to be more realistic, following the reviewer 2 comments. We think the new results are much better and explore the model skill better.
* * *
*P17, l491: WAIS growth is concerning, especially when exploring sea level change.*

**Response:** Yes, this a limitation of the model and we do point this out. Further development can potentially address this, as we discuss in the summary section. However, for the main aim of the GREB-ISM, which is 100kys global ice age cycle, the Northern Hemisphere continental ice sheets are much more important in driving global sea level changes. Our updated simulations show good agreement with the growth of these Northern Hemisphere ice sheets (see figure 10 and 11).
* * *
*P17, l506: I suggest removing this sentence especially since the next sentence refers to results showing all 3 oscillation periods.*

**Response:** We think the sentences need to stay, as the following paragraphs indeed focus on the 20kyrs period. This is then followed by paragraphs explaining how things are changing for the longer periods.
* * *
*P18, l536: this might be a resolution effect even though these places have been covered by ice sheets at some times.*

**Response:** It is interesting that these shallow ocean regions are ice sheet covered in our idealized experiments and as the reviewer points out this may indeed have happened in the past. We made no changes in response to this comment.
* * *
*P19, l566: replace"ISM-GREB" by "GREB-ISM" to be consistent.*

**Response:** It has been corrected.
* * *
*Tables*

*Table 5: Please define the term "boundary calving". Use proper citation for BedMachine (author and year instead of "BedMachine"). I could not find the Martin et al. (2011) in your references, please add it. Also, Martin et al. (2011) cites observation and does not derive it. Please correct your table accordingly.*

**Response:** It has been changed.
* * *
*Figures*

*Fig1: The green color scale encompasses too many layers and it is difficult to distinguish each of them. Please use an appropriate color scale. The blue scale is similar. Also, it is misleading*

*to use white for thickness lower than 300m. Right now it looks like white=no ice. Also, please improve the figure caption. You are not only showing the topography but also ice thickness.*

**Response:** It has been changed.

*Fig3: This figure needs to be improved. It is really difficult to compare it with Fig1 due to the color scale and the range of the colorbar. Please use "ice thickness" or "ice surface elevation" instead of "ice height" which can be misleading. Same remark as in Fig1 for the color scale. Please, distinguish in your figure what is sea ice to what is ice sheet.*

**Response:** Please, note Fig. 3 should not be compared to Fig. 1, as Fig 3. focuses on seasonal snow and sea ice cover, but Fig. 1 is focusing on large ice sheets. We did revise this figure to improve the color scheme.

*Fig4: Which year does panel (d) refer to? Improve the description of the figure in the caption. In particular, add what the black box is (it is cumbersome to go back and forth between figure and text to piece all the information together). Same remark as in Fig1 for the color scale.*

**Response:** Panel d is at the end of year 1. This is now mentioned in the figure caption. The color scheme in this picture is mostly distinguishable so we do small modification.

*Fig5: Please, indicate in the figure caption that the ice lifting takes place in the black box. Also, here or in the text, can you comment on why the response in Tsurf is limited to the black box? Modify the color scale for panel (a).*

**Response:** The figure has been revised. The Tsurf is limited to the black box is because cooling due to lifting does not lead to strong heat transportation to the surrounding. The potential air temperature, which is used to calculate the heat transport, in lifted region does not significantly change even though the surface temperature change. And air column at the point becomes shorter and thus it is difficult to be transported. Both effects reduce the potential heat transport because of lifting.

*Fig6: Define what "R" is in the caption.*

**Response:** It has been added.
* * *
*Fig8: Same remark for the colorbar as in Fig1 for the yellow and red here.*

**Response:** The figure has been revised. Please note, we also changed the grid for the simulations, to only show points where the simulation has ice sheets. The previous version did show velocities at grid points that did not have ice, as the velocity gird is different from the ice height grid.
* * *
*Fig9: panels (a,b) Focus on showing Greenland only if you are not showing results for other Northern Hemisphere ice growth (similarly to fig8 (a,b)). For Antarctica, show the full range of positive ice thickness; its extend does not match the one from the velocity plot and it is confusing at first. Increase the font of the colorbar label. Regarding the colors, see remark as for fig1.*

**Response:** We did revised the figure, but we keep showing the Northern Hemisphere. In this experiment ice can form outside of Greenland and it is therefore important to show all of the Northern Hemisphere.
* * *
*Fig11. Greenland looks quite small at -127ka. Can you comment in the text about it? I can't read the numbers on the colorbar, please increase the font size. Same remark for the color scale as in fig1.*

**Response:** We have changed this experiment and the results have changed accordingly. Please see the revised discussion.
* * *
*Fig 13: same remarks as in fig9.*

**Response:** We did revise the figure, but we keep showing the Northern Hemisphere. In this experiment ice can form outside of Greenland and it is therefore important to show all of the

Northern Hemisphere.
* * *
*Fig14: Increase the axis title font to be the same as axis tick labels. Increase the font size of the labels as well. Indicate in the caption that the normalization is done with respect to the last control time slice. Also, in the caption you write "orange, unit: 2W m-2" while the label reads 4W m-2. Which one is it?*

**Response:** We did revise the figure. There is no normalization done in this figure.
* * *
*Fig15: Use bigger font for label and axis title. In figure caption, remove "that". How does the ice volume increase with negative precipitation over 25kyr and 50kyr in panel (b) and (c) respectively?*

**Response:** We revised the figure following this advice.
* * *
*Fig16: same remark as in figure 1 for color scale.*

**Response:** We did revise the figure. Please note we also changed the way we present the ice sheet extent. In the previous version we only showed ice covered regions with ice sheets larger than 10m to focus on the ice sheets and not the snow/sea ice cover. However, this was not mentioned in the text and it did not fit well with the discussion. We now show the full ice cover including snow and sea ice cover.
* * *
*Fig17: increase the font size of color bar labels.*

**Response:** We made these revisions.

*The authors introduce a newly developed global ice sheet-climate coupled model which aims to explore the evolutions and variabilities of ice sheet-climate system on very long time scale. They first describe several changes applied in the GREB climate model and then introduce a new ice sheet model. Several suits of test experiments are performed to assess the ability of the model in simulating responses of ice sheet-climate system to idealistic forcing. They conclude that the model reproduces the general behavior of ice sheet variabilities reasonably well and state that it can be used for studies on very long time-scale, such as ice sheet-climate evolution over the Quaternary.*

*The manuscript is well written, easy to follow and its content matches the concept of GMD. In addition, as a paleoclimate modeller using comprehensive AOGCMs, I find this study and model quite interesting since it enables the community to explore the interactions and variabilities of ice sheet and climate in a global scale. On the other hand, I also find several concerns in the model as well as ambiguities in the aim of sensitivity experiments. For these reasons, I recommend major revision before accepting the paper for publication. Below summaries my main concerns, followed by specific comments.*

**Response:** We like to thank the referee for the evaluation of our manuscript and the many comments that will help us to improve the manuscript. Our one-by-one response is listed below.
* * *
*Major comments:*

*1. My main concern appears in the ambiguity on what kind of insights could the model provide in future studies. This model is clearly not meant to reproduce the ice sheet and climate precisely, but rather meant to reproduce them crudely so that the authors can discuss their interactions on global and long time scale. However, there is still a big problem in the spatial distribution of the simulated ice sheet; thickest glacial ice sheet exists over Siberia (Figs. 10 and 16). Previous studies have shown that the longitudinal locations of ice sheets affect their dynamical characteristics, which is important for interpreting the evolution of glacial-*

*interglacial cycle (e.g. Abe-Ouchi et al. 2013, nature, Fig.2b). Given that, how reliable a model with such a bias could be in discussing dynamics and evolutions of ice sheet and climate (e.g. glacial-interglacial cycle across the MPT, the effects of regolith on the ice sheet and so on).Therefore, I strongly encourage the authors to 1. put more efforts to reduce the thickest ice sheet in Siberia and 2. to discuss on what kind of insight could the model provide in understanding the ice sheet-climate system of the Quaternary.*

**Response:** The reviewers first point mentions several aspects. The first point is the ice sheet of Siberia. Please, note that this Siberian ice sheet results from an idealized model simulation that does not necessarily intend to reproduce past ice sheets. Following the reviewer's suggestions we changed the transient test experiments to follow the setup of Niu et al., (2019) (see also comments on point 3 below). After introducing the new boundary condition, the ice sheet distribution in LGM becomes similar to the result in other studies (Clark et al., 2009; Fyke et al., 2011; Ganopolski et al., 2010; Niu et al., 2019; Velichko et al., 1997; Willeit and Ganopolski, 2018) and the Siberian ice sheet disappears. It is able to capture large European (e.g. Fennoscandia) and North American (Laurentide) ice sheets.

The other oscillation simulations do also create a Siberian ice sheet, but they are idealized short wave forcing simulations that should only illustrate the capability of the GREB-ISM to respond to external forcing in a fully coupled climate system. Future applications in more realistic setups are needed to address this interesting problem. In the final section we highlight what kind of insights the new GREB-ISM model can give.
* * *
*2. While the ice sheet model simulates the ice shelf component of the ice sheet, I couldn't find discussions on the effect of basal melting at ice shelf-ocean margin. Perhaps this effect is partly incorporated through sea ice dynamics, but the authors should discuss possible effects of not including ice shelf basal melting in the model, especially over Antarctica. I assume this is partly causing the too thick west Antarctica in modern climate. Also, it might be related to the small sensitivity of Antarctic ice sheet to climate changes (L466-467).*

**Response:** We have indeed not included an explicit basal melting scheme but did test it. This is now discussed in the manuscript. The ice shelf velocity in our simulation is parametrized with a relatively large value for viscosity, this implicitly includes some other unresolved physical processes like basal melting effects. We have tested several basal melt schemes but the results do not have a fundamental improvement to our simulations.

For example, we applied an ice shelf basal melting scheme based on Martin et al. (2011):

$$\Delta H = \frac{\rho_o c_p \gamma_T F_{melt}(\Delta T_{add} + \beta H)}{L_m \rho_i}$$

Where $\Delta H$ is basal mass balance, $\gamma_T$, $F_{melt}$ are thermal exchange rate ($10^{-4}$) and model tuning factor ($5 \times 10^{-3}$), respectively. All of the other parameters can be found in Table 1. $\Delta T_{add} + \beta H$ in the equation is used to represent temperature difference between ocean and ice sheet basal melt point. This temperature difference is assumed to be a constant offset temperature $\Delta T_{add}$ plus a basal melting temperature modification based on ice thickness $H$.

Fig. R2 shows the dynamic equilibrium simulation after including basal melting scheme with temperature offset $\Delta T_{add}$ from 0 to 2K. The result indicates the basal melt scheme mainly reduces the ice thickness over Ronne ice shelf and part of Ross ice shelf. Its effect on WAIS is very limited. Consequently, we do not include an extra basal melt scheme in our model. We discuss other possible explanations for the unrealistic WAIS ice thicknesses in section 6 as well.

[Figure]

**Figure R2.** Ice thickness difference [m] between models with ice shelf basal melting and the GREB-ISM current version in forced dynamic equilibrium experiments.
* * *
*3. The aim of transient test experiments of ice sheet is unclear to me (L434-L437). The authors apply a reconstructed Greenland temperature used in Greve (1997) to their ice sheet model over the entire northern high latitudes. This method may be appropriate for Greve (1997), which focused on Greenland ice sheet, or a 2D (latitude and height) ice sheet model. However, it is clearly not suitable for a study focusing on 3D Northern Hemisphere ice sheets, since it overestimates/underestimates glacial-interglacial temperature differences outside the Greenland area (e.g. Kageyama et al. 2021, Climate of the Past, Fig. 2b, https://doi.org/10.5194/cp-17-1065-2021). Partly as a result, the thickest ice sheet develops over Siberia, which is affecting the simulated sea-level change. I would recommend the*

*authors to use different methods such as that used in Niu et al. (2019, Journal of Glaciology, doi: 10.1017/jog.2019.42). In this study, they force the ice sheet model with a similar index method, but utilise information from climate models in considering the regional dependences in temperature changes. In this method, one can assess whether the ice sheet model can simulate the geographical location and volume of ice correctly when reasonable atmospheric forcing is applied.*

**Response:** As suggested by the reviewer, we have adopted the forcing from Niu et al., (2019) and rerun the transition experiment. This change does improve our simulation and removes the growth of a Siberian ice sheet. We have modified the transition experiment part in our text.
* * *
*4. Metrics of assessing model's reproducibility are sometimes vague to me. For example, the authors state their model generally reproduces the ice velocity (Fig. 8 & L432) and sea level change reasonably well, but I couldn't find how they defined that (sorry if I missed). Is it because the model reproduces the order of magnitude correctly? Please add an explanation on this.*

**Response:** We have now provided a better discussion of how the simulated velocities match the observed in Section 4.3. We further changed the transient experiments by using the forcing from Niu et al., (2019) and the resulting sea level changes are now more similar to those observed.
* * *
*Specific comments*

*L30: "procession" -> "precession"?*

**Response:** It has been corrected.
* * *
*L53: Perhaps you may also cite a recent study by Willeit et al. (2019, Science advances, eaav7337) here.*

**Response:** It has been added.
* * *
*L91 and Table 1: I don't understand what "air thermal heat" means. Could you add an explanation in the main text or table?*

**Response:** It refers to the net longwave radiation for the atmospheric temperature. It has been changed in Table 1.
* * *
*L99: It may be useful to add a sentence describing how you estimated the correction terms for temperatures. (Sorry if I missed them)*

**Response:** Done (section 3.1).
* * *
*L140: Is there any reason behind the choice of 10 m threshold? If so, please add a short explanation.*

**Response:** Yes, we assume that ice < 10m does not flow. This is mostly seasonal snow cover and sea ice (Fyke et al., 2011). To make the point clearer, we added a sentence in the text in section 3.2.2.
* * *
*L168: What is Lice? Is it explained in the manuscript or is it a typo of Lm?*

**Response:** It is a typo of Lm and has been corrected.
* * *
*L180 and Fig.3: How does the simulated ice cover (or thickness) compare to modern observation? This tells the biases in the model, which is important to know when interpreting the evolution of the ice-climate system of the past.*

**Response:** This section focuses on the seasonal ice cover. Both the land snow cover and ocean sea ice distribution compare well with observed values (Rayner et al., 2003; Robinson et al., 2012). The section has been revised to better discuss this.
* * *
*L272-273: Making a bold assumption is sometimes inevitable in relatively simple models, but the authors should present some evidences supporting this assumption.*

**Response:** We have reframed this section and better explained how we developed this approach.
* * *
*L288 and equation (32): Why do you set a maximum ice thickness for ice growth? Please add a sentence on it.*

**Response:** We have added an explanation for this. The sea ice growth threshold of 0.5 m is reflecting the fact that sea ice is a very good insulator and subsequently does not transfer atmospheric heat fluxes very well once a certain ice thickness is reached. This in practice limits the growth of sea ice to less than a 1m typically.
* * *
*L379: Would be helpful for readers if you refer to Table 3 here.*

**Response:** Done.
* * *
*L398: Do you mean Experiment B is warmer by 5K compared to Experiment A?*

**Response:** Experiment B is cooler by 5K compared to Experiment A. The sentence is corrected.
* * *
*L423: Isn't it simply because your model does not account for strong ice shelf basal melting in West Antarctica? Perhaps you can crudely account for this effect by increasing the calving in this area.*

**Response:** We have tested several basal melting schemes but it does not solve the issue. Calving rate may be a solution for the issue. However, increasing calving rate also changes the ice thickness in other places with good model performance, such as Greenland. We may be able to "fine tune" this for West Antarctica, but we have not done this. We have revised our discussion of this issue and also add some discussion in the final section.
* * *
*L447: As far as I know, Greenland ice core data extends back to 130ka. How did you force the ice sheet model before this period. Perhaps, it may be written in Greve (1997), but please add a sentence on this point.*

**Response:** Actually, we directly obtained the data from Ralf Greve. The Greenland Ice Core Project (GRIP) data link is https://www.ncei.noaa.gov/access/paleo-search/study/17839. The data spins from -248 kyr to -39 yr. To help reader better understand the data, we have added the original data citation Johnsen et al., (1997) in our data description part.
* * *
*L453: Do you have any possible explanation why the model underestimate the amplitude of the sea level variability? A sentence would be sufficient.*

**Response:** We used the new equation (41) to define the climate forcing for the transition experiment, which is similar to Niu et al., (2019). After changing the setup of this transition experiment substantially, the new sea level change is around 120 m, which is very close to the literature.
* * *
*L459: "EIVM", do you mean by LIG here?*

**Response:** Yes, it is LIG. It has been corrected.
* * *
*L465: Do you mean "our simulation of"?*

**Response:** It has been corrected.
* * *
*L516: Why does the mean ice volume increase after the first cycle?*

**Response:** We added another paragraph to further explain this effect. It is related to the fact that the control mean ice cover is already very small and cannot decrease much further for increased SW forcing. However, it can grow much larger for decrease SW forcings.
* * *
*L564: Could you further elaborate on how the coarse resolution of ice sheet model causes the bias in West Antarctica?*

**Response:** Large area of ice sheet in West Antarctica and Antarctica Peninsula is marine ice sheet, which is very sensitive to climate forcing, topography and grounding line dynamics. Simulation of those elements are all related to the model resolution. We added a sentence explaining that the complex topography may play a role here.
* * *
*L567-568: Since the authors use Greenland temperature record to force the ice sheet model, I assume that the effect of changes in ocean circulation on ice sheets are already partly incorporated. Could you further elaborate on how the absence of ocean circulation affects the the biases in simulated North American and Eurasian ice sheet?*

**Response:** We have now revised this simulation setup, using new boundary condition from Niu et al., (2019). This experiment now has incorporated all surface temperature changes from a Coupled Global Climate Model simulation forced by insolation, greenhouse gas and ice sheet. Thus, it does include ocean circulation changes. A discussion has been included in Section 4.4.

---

## Referee Report (RR1)

**Review of Xie et al. 2021, GMD**

The comments I made have been mostly addressed and I found this version of the manuscript easier to follow. The figures are greatly improved and much easier to read.

At this point I only have a few minor suggestions and one clarification to request after which the study would be ready for publication. The lines and page numbers refer to the latest version of the manuscript.

P5, l127: replace "thickness and temperature" by "thickness, velocity, and temperature". Otherwise, it reads that you are solving 3 equations for 2 unknowns while you have section 3.2.4 that shows the SSA equation.

P8, Sec 3.2.3.1: Thanks for adding this section. What I understand from it is that ice in the floating or grounded ice mask will need to thin down to less than 10m to become part of the ocean mask which then would become sea ice. If I am understanding this wrong, please add some details to this section. Also, please add a couple of sentences to describe what is happening to this ice that is now part of the ocean mask but has a thickness greater than 0.5 m. Later in the text, you mention that sea ice is not allowed to grow more than 0.5 m. Does this mean you quickly melt this thicker ice to be less than 0.5m? Please clarify this aspect here.

P9, l239: what is the value of the constant viscosity?

P17, l472 and 482: The SSA is an approximation rather than a parameterization. (It approximates the Stokes equation.)

P17, l480: To be clear, BedMachine is a product for bed topography and ice thickness amongst other variables, but not velocity. Please remove its reference here.

P17, l482: "which is due to the parameterization… " I would leave this part of the sentence out or tone it down with a word likes "partly". You have not explored many options to pinpoint "THE" reason of your velocity mismatch. As I mentioned in my first round of comments resolution is one of them, but your constraint on ice viscosity could be another one and your choice of inversion process.

P18, l505: replace "0.67" by "-0.67", whatever is correct value between here and Fig. 10.

Fig 8: replace "Obsevation" by "Observation" in the title of the left column.

---

## Referee Report (RR2)

In the revised manuscript, the authors mostly addressed my concerns and questions raised in the previous review. In particular, they conducted additional sensitivity experiments similar to Niu et al. (2019) and showed that their new ice sheet model is capable of simulating changes in ice sheet over the glacial-interglacial cycle when reasonable atmospheric forcing is applied. On the other hand, I couldn't find sentences in the last section highlighting what kind of insight can the model give to better understand the ice-age cycle over the Quaternary. Note that in the response letter, it did say the authors included those sentences (page 20). I did find several interesting sentences in the track-trace file that attempt to make this point, but those sentences were deleted. Does this file really compare the most recent manuscript with the one that was published in the Discussion paper? Given this condition, I cannot make a decision at the moment. Below shows some comments that may help to improve the manuscript.

L408-409: The latter part of the sentence is inaccurate since the sensitivity experiment is forced with a combination of Greenland isotope record and model outputs from PMIP. Please modify it.

L423-424: It would be better to clarify that the authors are discussing the result of moving margin experiment here.

L428: Sorry if I have misunderstand, but I thought the authors are discussing the results from transient experiments in Fig. 6. If so, why is it citing Table 4, which shows result of steady state experiments?

L497-499: I assume the authors are using only the raw LGM climate condition from AWI climate simulations. In that case, it would be better to just refer to $T_{LGM}(\lambda,\phi,t_{day})$. $((T_{LGM}(\lambda,\phi,t_{day}) - T_{today}(\lambda,\phi,t_{day}))$ gives an impression the the ice sheet model is forced with anomalies between LGM and piControl in AWI model. However, that is not the case, right?

---

## Author Response (AR2)

**Revisions of "GREB-ISM v1.0: A coupled ice sheet model for the Global Resolved Energy Balance model for global simulations on time-scales of 100 kyr"**

Dear Editor and referees,

We would like to thank the two referees and the editor for their time spent on reviewing this manuscript, and for the many very helpful comments they provided. We think the referee comments have helped us to substantially improve the presentation of this work. Below we give a point-to-point response to all referee comments, hoping the revised manuscript has now been improved in clarity and is ready for publication.

With best regards,

Zhiang Xie, Dietmar Dommenget, Felicity S. McCormack, and Andrew N. Mackintosh

The comments I made have been mostly addressed and I found this version of the manuscript easier to follow. The figures are greatly improved and much easier to read.

At this point I only have a few minor suggestions and one clarification to request after which the study would be ready for publication. The lines and page numbers refer to the latest version of the manuscript.

**Response:** We like to thank the reviewer again for the informative feedback, which helped improve our manuscript a lot. The detailed response is listed below.
* * *
P5, l127: replace "thickness and temperature" by "thickness, velocity, and temperature". Otherwise, it reads that you are solving 3 equations for 2 unknowns while you have section 3.2.4 that shows the SSA equation.

**Response:** We revised this sentence now state that we have two prognostic equations and the additional diagnostic velocities.
* * *
P8, Sec 3.2.3.1: Thanks for adding this section. What I understand from it is that ice in the floating or grounded ice mask will need to thin down to less than 10m to become part of the ocean mask which then would become sea ice. If I am understanding this wrong, please add some details to this section. Also, please add a couple of sentences to describe what is happening to this ice that is now part of the ocean mask but has a thickness greater than 0.5 m. Later in the text, you mention that sea ice is not allowed to grow more than 0.5 m. Does this mean you quickly melt this thicker ice to be less than 0.5m? Please clarify this aspect here.

**Response:** The reviewer is correct, floating ice comes ocean grid point if it is < 10m. In Section 3.3.4 we explain how sea ice tendencies are computed. Here sea ice can be larger then 0.5m, but it cannot grow larger than 0.5m by atmospheric heat fluxes. The Sea ice transport by

isotropic diffusion ($\kappa_{si} \nabla^2 H$) will reduce the 10m sea ice fast. So we are not quickly melting the sea ice, but it will be diffused by transport. We added some texts to highlight that only atmospheric heat fluxes is limited.
* * *
P9, l239: what is the value of the constant viscosity?

**Response:** It is in Table 1. The table has been cited now.
* * *
P17, l472 and 482: The SSA is an approximation rather than a parameterization. (It approximates the Stokes equation.)

**Response:** The parameterization we mentioned here is referred as to the viscosity parameterization for SSA.
* * *
P17, l480: To be clear, BedMachine is a product for bed topography and ice thickness amongst other variables, but not velocity. Please remove its reference here.

**Response:** It has been corrected now. We used MEaSUREs data as mentioned in section 2.
* * *
P17, l482: "which is due to the parameterization… " I would leave this part of the sentence out or tone it down with a word likes "partly". You have not explored many options to pinpoint "THE" reason of your velocity mismatch. As I mentioned in my first round of comments resolution is one of them, but your constraint on ice viscosity could be another one and your choice of inversion process.

**Response:** We revised the expression.

P18, l505: replace "0.67" by "-0.67", whatever is correct value between here and Fig. 10.

**Response:** Sorry, this is typo in the text. It is -0.67. To make a better comparison, $\delta^{18}O$ proxy data axis has been inverted in Fig. 10 and we now clarify it in the caption.

Fig 8: replace "Obsevation" by "Observation" in the title of the left column.

**Response:** It has been corrected.

*Referee #2*

Review of Zhiang et al.

In the revised manuscript, the authors mostly addressed my concerns and questions raised in the previous review. In particular, they conducted additional sensitivity experiments similar to Niu et al. (2019) and showed that their new ice sheet model is capable of simulating changes in ice sheet over the glacial-interglacial cycle when reasonable atmospheric forcing is applied. On the other hand, I couldn't find sentences in the last section highlighting what kind of insight can the model give to better understand the ice-age cycle over the Quaternary. Note that in the response letter, it did say the authors included those sentences (page 20). I did find several interesting sentences in the track-trace file that attempt to make this point, but those sentences were deleted. Does this file really compare the most recent manuscript with the one that was published in the Discussion paper? Given this condition, I cannot make a decision at the moment. Below shows some comments that may help to improve the manuscript.

**Response:** Thanks for your comments. The uploaded version was the final version. We deleted those sentences since we realised the summary part has covered the main points of our model's new insight: global scale and high model efficiency. So, we think it has been highlighted already in the last section. However, considering your comments, we reinclude the detailed discussion about those new insights in the final section. We hope the current version does illustrate the advantage of the GREB-ISM.

L408-409: The latter part of the sentence is inaccurate since the sensitivity experiment is forced with a combination of Greenland isotope record and model outputs from PMIP. Please modify it.

**Response:** It has been corrected by citing Niu et al., (2019).

L423-424: It would be better to clarify that the authors are discussing the result of moving margin experiment here.

**Response:** Done.
* * *
L428: Sorry if I have misunderstand, but I thought the authors are discussing the results from transient experiments in Fig. 6. If so, why is it citing Table 4, which shows result of steady state experiments?

**Response:** Yes, the reviewer is right. We are discussing the transient experiment. Here we should cite Fig. 6 instead of Table. 4. This has now been corrected.
* * *
L497-499: I assume the authors are using only the raw LGM climate condition from AWI climate simulations. In that case, it would be better to just refer to TLGM($\lambda$,$\phi$,$tday$). ((TLGM($\lambda$,$\phi$,$tday$) −Ttoday($\lambda$,$\phi$,$tday$)) gives an impression the the ice sheet model is forced with anomalies between LGM and piControl in AWI model. However, that is not the case, right?

**Response:** Correct. Our LGM climatology is from AWI climate simulation while variables at present are from observation. We have modified the corresponding part in the text.